# Water Deficit Stress Tolerance Potential of Newly Developed Wheat Genotypes for Better Yield Based on Agronomic Traits and Stress Tolerance Indices: Physio-Biochemical Responses, Lipid Peroxidation and Antioxidative Defense Mechanism

**DOI:** 10.3390/plants11030466

**Published:** 2022-02-08

**Authors:** Sumreena Shahid, Qasim Ali, Shafaqat Ali, Fahad A. Al-Misned, Saliha Maqbool

**Affiliations:** 1Department of Botany, Government College University, Faisalabad 38000, Pakistan; sumreenashahid1992@gmail.com; 2Department of Environmental Sciences and Engineering, Government College University, Allama Iqbal Road, Faisalabad 38000, Pakistan; 3Department of Biological Sciences and Technology, China Medical University, Taichung 40402, Taiwan; 4Department of Zoology, College of Science, King Saud University, Riyadh 11451, Saudi Arabia; almisned@ksu.edu.sa; 5Department of Soil, Water, and Climate, University of Minnesota, Saint Paul, MN 55108, USA; smaqbool@umn.edu

**Keywords:** water stress, breeding, agronomic traits, yield, oxidative stress, water relations, photosynthetic pigments

## Abstract

Changing environmental conditions, fresh water shortages for irrigation and the rapid increase in world population have created the problems of food insecurity and malnutrition. Different strategies, including the development of water stress-tolerant, high-yielding genotypes through breeding are used to fulfil the world food demand. The present study was conducted for the selection of high-yielding, drought-tolerant wheat genotypes, considering different morpho-physio-biochemical, agronomic and yield attributes in relation to the stress tolerance indices (STI). The experiment was carried out in field in a split-plot arrangement. Water deficit stress was maintained based on the number of irrigations. All genotypes showed a differential decreasing trend in different agronomic traits. However, the increasing or decreasing trend in leaf photosynthetic pigments, non-enzymatic and enzymatic antioxidants under limited water supply also found to be genotype-specific. Genotypes MP1, MP3, MP5, MP8 and MP10 performed better regarding the yield performance under water deficit stress, which was associated with their better maintenance of water relations, photosynthetic pigments and antioxidative defense mechanisms. In conclusion, the physio-biochemical mechanisms should also be considered as the part of breeding programs for the selection of stress-tolerant genotypes, along with agronomic traits, in wheat.

## 1. Introduction

The shortage of fresh water for irrigation, combined with the ever-increasing population, is a major hindrance to fulfilling world food demand. It is expected that this situation will become worse for researchers in the near future due to the changing environment, as well as patterns of rainfall that have further increased the aridity problem in many areas of world [1]. Agriculturalists are facing the problem of, by the middle of 21st century, meeting the requirements for the food demands of nine billion people [2]. The production of more food with less availability of water, especially in semi-arid and arid zones, is also an emerging challenge for agriculturalists nowadays [3]. Different agricultural areas of the world are facing the problem of land degradation due to the shortage and scarcity of water [4,5]. A total of 80% of the world’s agriculture depends on rain-fed land, which produces 80% of the food globally [6,7]. Approximately 95% of land in North Africa and West Asia is rain-fed, and 40% of the land in Uzbekistan faces water shortages, causing the problem of despoiled fields [8,9]. In Pakistan, approximately 12.53 million hectares are irrigated with underground water, 6.35 million hectares of agriculture land are cultivated with canal water, and no water is available for the remaining 3.59 million hectares, from a total of 22.45 million hectares [10,11].

Under such challenging conditions, to fulfill the demand for food at a global level, it is either necessary to increase the cultivated land or to produce high-yielding stress-tolerant crop varieties. The former option seems unfeasible due to limitations of fresh water for irrigation. In crop plants, the stress tolerance is the function of alteration in different physiological and biochemical mechanisms, which leads to clear morphological adaptation [12]. These mechanisms are better water content, the increased accumulation of photosynthetic pigments leading to a better photosynthetic activity, and efficient antioxidative defense mechanisms (enzymatic and non-enzymatic antioxidants). Different physiological and biochemical changes are studied in agronomic plants including the bread wheat and found genotype-specific in their responses to drought stress [13,14,15,16]. The plant species with better performances in all (or a few) of the above-mentioned mechanism are categorized as drought-sensitive or tolerant species. The better performance in the yield of crop plants, especially under adverse environmental conditions, is the function of physio-biochemical mechanisms.

The physio-biochemical traits of plants are considered an important selection tool under stressful environmental conditions, especially water deficit stress, because they have a direct relationship with the plant adaptations to stressful conditions [17]. Plant physio-biochemical responses under water deficit stress include a decreased photosynthetic activity due to membrane disruptions by oxidative stress, the accumulation of toxic metabolites, and perturbations in cellular water relations, leading to a loss of turgidity, and reduction in growth rates and productivity. It is reported that, in crop plants, different physiological responses are linked with their resistance functions under drought, such as a high amount of relative water content, water potential and integrity of membranes [18,19,20]. Perturbations in biomass production are considered as one such important process which is sensitive to drought and linked with the disturbances in cellular water relations [21].

The performance of crop plants regarding physio-biochemical mechanisms is responsible for survival under drought; that are found plant species and/or cultivar-specific [12]. Furthermore, the mechanism for plant survival also changes under different environmental conditions that clearly adhere to the genetic potential of plants [22]. In this regard, the selection of high-yielding crop varieties with rapidly changing conditions of environment is considered a complex mechanism and plant-species-specific [18]. In the case of cereals, regarding to that of wheat that is hexaploid in nature, the stress tolerance mechanism likely becomes more complex in exploring and understanding. Therefore, to address the present challenging environmental conditions, the selection from available germplasms, or the development of new cultivars through breeding or genetic engineering for stress tolerance, trait selection is indeed of prime importance to fulfill future world food demand [12].

Mostly in crop plants, their selection for better production under drought is mainly based on the different agronomic traits, including biomass and yield traits, and very few cases are reported, considering the different physiological and biochemical attributes in parallel with agronomic traits, which are the main enigma behind the stress tolerance of crop plants [12]. Though the agronomic traits are considered as the most important traits in field crops for their selection for better production under drought [12], it is necessary to select the cultivars with a better yield under stressful environments by considering the physiological and biochemical traits, along with agronomic traits. Work is already in progress regarding the aforementioned aspects, but the mechanism is still unclear. The world’s most major food dependence is on cereals, especially wheat, including in Pakistan where wheat has a complex mechanism due to its hexaploidy nature. Pakistan is an agricultural country and 70% of its population is dependent on agricultural crops for food. Being a staple food, wheat is most consumed crop among the important cereal crops and helps to solve food security problems. Pakistan is in sixth position worldwide for global wheat production and in eighth position for cultivation, producing approximately 3.5% wheat annually [23,24].

World demands more food due to a fast increase in human population and is expected that it will reach to 40% by 2030. Therefore, crop production must be increased to meet increased food demands and attain the goals of sustainable food security [12,25]. The global population is estimated to reach 9.8 billion people by 2050 [26]. Therefore, there is a dire need to increase crop productivity by 70% to fulfill the projected demand by the middle of the 21st century [27]. In this regard, the major focus of the breeders is to increase the productivity of wheat at a global level as it faces the serious threat of ongoing and increasing adverse environmental conditions. Due the various abiotic stress constraints, including drought, wheat production was compromised [19,28]. Therefore, the primary goal of wheat breeders is developing drought-tolerant wheat cultivars.

Wheat is one of the most important founder crops in agricultural, started about 10,000 years ago in Fertile Crescent. It is developed through long evolutionary process, involving the different ancestors [26,27,29]. The genetics of hexaploidy wheat is very complicated due of its polyploidy. It results into genes duplication and triplication that leads to complex epistatic effects and segregational patterns that are difficult to analyse Sentenseand to resolve the effects of component genes. Therefore, the genes description and location in wheat lags is similar to diploid crop barley [30]. Therefore, the objectives of the study were the selection of drought-tolerant wheat genotypes from a set of experimental lines developed from a cross of high-yielding, stress-tolerant wheat varieties based on the agronomic traits, such as plant biomass production and yield attributes, in relation to biochemical and physiological traits, including leaf photosynthetic pigments, water relations of plant, lipid peroxidation and antioxidative defense mechanisms. Moreover, the studies of stress tolerance indices were also under consideration.

## 2. Materials and Methods

The experiment was conducted in a field research area of Botanical Garden (latitude 30°30′ N, longitude 73°10′ E and altitude 213 m) of Government College University, Faisalabad in the two consecutive years during the 2018–2019 and 2019–2020 wheat growth seasons. The repetition of the experiment was conducted for the confirmation of results. Experimental design was a split plot. Selected wheat lines developed from a cross (Millat ♂ × Punjab ♀) were grown along with the parental varieties. A total of 12 genotypes (10 inbreed lines selected from the cross with the two parental varieties) were sown for experiment. Research area allocated to the experiment comprised two main plots corresponding to each irrigation level (water deficit stress and normal irrigation). Each main plot was divided into 12 subplots (6 × 25 ft^2^ each) corresponding to specific wheat genotype. There were 4 lines in each subplot with a row-to-row distance of 15 cm. The plot allocated to normal irrigation was watered as per the irrigation requirements. While the water-deficit stressed plots were watered twice during the experimental period (at seedling and spike initiation stages, respectively). The field was well-prepared before seed sowing, by irrigating the soil with canal water. When the soil was at field capacity after 15 days of irrigation, the soil was well ploughed for sowing of seeds. After ploughing the seeds were hand sown with 15 cm row-to-row distance. After 3 weeks of seed sowing, the first irrigation was applied to both main plots. The thinning was conducted manually after 15 days of the 1st irrigation with 10 cm plant-to-plant distance. An adequate amount of K, P and N fertilizers were given in soil as per the recommendations.

### 2.1. Soil Physico-Chemical Properties

Davis and Freitas [31] assay was followed for studying the chemical and physical characteristics of soil. The soil of the experimental site was sandy loam in texture, with an available total organic matter (1.15%), P (8.6 ppm), and N (0.73%) with a saturation percentage of 34%. The average of EC and pH of experimental soil was 253 ds.m^−1^ and 7.8, respectively. The soil solution had the SAR (0.086 meq L^−1^), Na^+^ (2.98 meq L^−1^), Fe (0.041 meqL^−1^), Ca^2+^ + Mg^2+^ (14.3 meq L^−1^), Cl^−^ (8.52 meq L^−1^), SO_4_^−2^ (1.98 meq L^−1^), soluble CO_3_^2−^ (traces) and HCO_3_^−^ (4.93 meqL^−1^).

#### Data Collection

Collection of the data for different morphological and physio-biochemical parameters was carried out after 15 days of the second irrigation, which was given to the water-stressed plots at the spike initiation stage, while the data for different yield attributes were collected at the maturity. The plots were covered with polythene sheets to avoid the effects of rainfall during the experiment. Flag leaf was used for the measurement of all biochemical and physiological parameters. For the measurements of morphological and growth parameters, two plants per replicate were used and separated into roots and shoots. After measuring the shoot and root fresh masses, the plant samples were dried at 70 °C for 48 hr for the estimation of dry masses of root and shoot. At the same growth stage for measuring the physio-biochemical attributes, the fresh leaf samples (only flag leaf) were taken and stored at −80 °C to be used later.

### 2.2. Climatic Conditions

During the experiment period, the averages of different climatic conditions such as temperature, relative humidity, and available radiation were recorded. The climatic conditions at the site, calculated as means, were as follows: the photosynthetically available radiation measured at noon varied from 501 to 906 µmol m^−2^ s^−1^, day and night temperatures between 12 ± 3.56 °C and 25 ± 2.75 °C, mean day/night RH 38.83/75.1%, and rain fall 27.11 mm, respectively.

### 2.3. Leaf Relative Water Content (LRWC)

Fully matured top leaf (flag leaf) was used for the estimation of LRWC. The leaves were cut with the help of scissors, and fresh weights (FW) were estimated. After marking with specific tags, the leaves were then dipped in water for 4 h. After 4 h, the turgid weight (TW) of leaves was measured. The dry weights (DW) of the leaves were then estimated after drying leaves at 70 °C for 48 h. The LRWC given below was then used for the estimation of LRWC:
$$\mathrm{LRWC}(\%)=\frac{\mathrm{Fresh}\;\mathrm{weight}\;\mathrm{of}\;\mathrm{leaf-Dry}\;\mathrm{weight}\;\mathrm{of}\;\mathrm{leaf}}{\mathrm{Turgid}\;\mathrm{weight}\;\mathrm{of}\;\mathrm{leaf-Dry}\;\mathrm{weight}\;\mathrm{of}\;\mathrm{leaf}}\times 100$$


### 2.4. Determination of Leaf Photosynthetic Pigments

The estimation of leaf Chl. *a*, Chl. *b*, total Chl. and Chl. *a/b* was carried out using the method of Arnon [32]. However, the method of Kirk and Allen [33] was followed for the carotenoid determination. The fresh leaves (0.1 g) were finely chopped (1 cm^2^ pieces), and 5 mL of acetone (80%) was used for extraction, kept for 24 h at 4 °C. The absorbance of the extract was measured at 663 nm, 645 nm and 480 nm using a spectrophotometer (Hitachi U-2001, Tokyo, Japan). The quantification of the photosynthetic pigments was estimated using formulas given below:T. Chl. = [20.2 (OD 645) − 8.02(OD 663)] × v/w × 1/1000
Chl. *a* = [12.7 (OD 663) − 2.69 (OD 645)] × v/1000 × w
Chl. *b* = [22.9 (OD 645) − 4.68 (OD 663)] × v/1000 × w
A Car. (µg/g FW) = OD 480 + (0.114 x OD 663) × (0.638 × OD 645)
Car = A Car/Em 100% × 100
Emission = Em 100% = 2500
OD = absorbance at respective wavelength
V = volume of the extract (mL)
W = weight of the fresh leaf tissue (g)

### 2.5. Determination of Leaf Relative Membrane Permeability (LRMP)

The LRMP was estimated following the method as reported by Yang et al. [34]. Fresh leaf material (0.5 g) was chopped and dipped in 20 mL of dH_2_O using test tubes. The test tubes were then vortexed for 5 s and EC_o_ was measured using an EC meter. These test tubes were then kept for 24 at 4 °C and EC_1_ was estimated. The samples containing test tubes were then autoclaved at 120 °C and the EC_2_ of the assayed material was estimated. The formula given below was then used for the quantification of LRMP (%):
$$\mathrm{LRMP}(\%)=\frac{{\mathrm{EC}}_{1}-{\mathrm{EC}}_{o}}{{\mathrm{EC}}_{2}-{\mathrm{EC}}_{o}}\times 100$$


### 2.6. Determination of Leaf Total Phenolic Contents (TPC)

TPC in leaf were determined using the method of Julkenen-Titto [35]. Fresh leaf extract prepared in 80% acetone was centrifuged for 10 min at 10,000× *g*. After the centrifugation (14,000× *g*), Folin–Ciocalteau’s phenol reagent (1 mL), dH_2_O (2 mL), and 5 mL of Na_2_CO_3_ (20%) solution were added to the obtained supernatant (100 µL). Then, dH_2_O was added to create the final volume of 10 mL. The absorbance was then read at 750 nm of the mixture using a UV-Visible spectrophotometer (IRMECO U2020) (GmbH, Geesthacht, Germany).

### 2.7. Estimation of Leaf Ascorbic Acid (AsA) Content

The estimation of AsA contents in leaf was carried out using the method ascribed by Mukherjee and Choudhuri [36]. Shortly after the preparation of reaction mixture, the OD of the reaction mixture was read at 530 nm, and the quantification of AsA in leaf samples was conducted using the standard curve prepared from standard solutions (50–300 ppm), using the pure standards.

### 2.8. Estimation of Leaf Flavonoids

The method ascribed by Sultana et al. [37] was followed for the determination of flavonoid content in leaf tissues by using the methanolic extract of leaf. To 1 mL of extract, 0.3 mL of 5% NaNO_2_ was added. After 6 min, 2 mL of 1 M NaOH and 2.8 mL of distilled water were added to the reaction mixture after incubating the resultant solution at the room temperature for 40 min. The OD of the mixture was read at 430 nm. A range of catechin standards were used for the quantification of flavonoids.

### 2.9. Determination of Leaf Total Anthocyanin Content

The estimation of leaf anthocyanin content was performed using 80% acidic methanol [38]. Shortly, after grinding fresh leaf material (0.25 g) in 10 mL acidic methanol and was centrifuged for 5 min at 14,000× *g*. The absorbance of the solution was read at 530 nm and 657 nm. The quantification of anthocyanin contents was conducted using the equation given below:Anthocyanin = A657 × (A530 − 0.25) × M^−1^
where A530 and A657 are the absorptions at the specific wavelengths, and M is the fresh leaf mass used for the extraction (g).

### 2.10. Estimation of Leaf Total Soluble Protein (TSP) and Enzymatic Antioxidants

#### 2.10.1. Extraction of Enzymatic Antioxidants and TSP

The extraction of enzymatic antioxidants and TSP was completed by grinding the 0.5 g fresh leaf material in 50 mM chilled phosphate buffer (10 mL). After centrifugation (14,000× *g*) at 4 °C for 5 min, the obtained supernatant was used to estimate the levels of TSP and enzymatic antioxidants.

#### 2.10.2. TSP Estimation

The method ascribed by Bradford [39] was followed for the estimation of TSP by using Bradford reagent. To the 100 µL of buffer extract 2 mL of Bradford reagent was added, mixed well, and the OD was read at 595 nm. The quantification of TSP was performed using a standard curve from a range of protein standards (200–14,000 mg/kg) prepared from bovine serum albumin of analytical grade.

#### 2.10.3. Determination of Superoxide Dismutase Activities (SOD)

The activity of SOD in leaf samples was determined by following the method as described by Giannopolitis and Ries [40]. For the preparation of reaction mixture, 13 mM methionine, 1.3 μM riboflavin, 75 nM EDTA, and 50 μM NBT were added to 50 μL enzyme extract prepared in 50 mM phosphate buffer (pH 7.8). The finally prepared reaction mixture was then subjected to a florescent light source for 15 min using a box coated with aluminum foil on the inner side. Finally, the OD of the reaction mixture was read at 560 nm. The reaction mixture without samples was taken as blank. The SOD activity in samples was estimated as unit/mg protein using the content of TSP.

#### 2.10.4. Estimation of Peroxidase Activity (POD)

The activity of POD was assayed following the method of Chance and Maehly [41] using leaf sample as prepared in phosphate buffer for the estimation of TSP and SOD. The oxidation of guaiacol was the basic mechanism behind the POD estimation. The reaction mixture was prepared by adding 40 mM H_2_O_2_, 100 μL leaf phosphate buffer extract, and 20 mM guaiacol to 50 mM phosphate buffer. The oxidation of guaiacol changed the color of the reaction mixture, and the reaction mixture absorbance was read after every 20 s at 470 nm for 180 s. The final POD activity was measured as unit/mg protein.

#### 2.10.5. Determination of Catalase (CAT) and Ascorbate per Oxidase (APX) Activities

The method of Chance and Maehly [41] was followed for the determination of CAT. The basic mechanism behind the estimation of CAT activity was based on the disappearance of H_2_O_2_ in the reaction mixture, and the absorbance of the reaction mixture was read at 240 nm after 20 s in time-scanning manners. The reaction mixture was prepared by adding 100 µL of leaf extract in 1 mL of dH_2_O, and 1.9 mL of H_2_O_2_. However, the measurement of APX activity was completed following the method of Asada and Takahashi [42]. The OD was read at 290 nm and calculated in units mg^−1^ protein.

### 2.11. Estimation of Leaf Total Free Amino Acid (FAA)

The method of Hamilton and Van Slyke [43] was used for the FAA estimation by using leaf samples prepared in buffer extract. One milliliter of leaf extract was added to 1 mL of ninhydrin (1%) and 1 mL of 2% pyridine, heated at 95 °C for 30 min, and made the volume up to 50 mL by adding dH_2_O. The OD of the reaction mixture was read at 570 nm. The FAA quantification was conducted using the equation below:Total FAA = Abs × V × DF/wt of sample × 1000

### 2.12. Estimation of Reducing Sugars (RS)

The method of Wood and Bhatt [44] was followed for the determination of RS using leaf sample (0.5 g) prepared in methanol solution (80%), after centrifuging at (14,000× *g*) at the room temperature. To 1 mL of reaction mixture, DNS (4 mL) was added, followed by heating at 95 °C for 5 min in a water bath. After cooling in ice-chilled water, the reaction mixture was then incubated at 25 °C, and the absorbance was read at 540 nm.

### 2.13. Determination of Non-Reducing Sugars (NRS) and Total Soluble Sugars (TSS)

The TSS was quantified using the method of [45] by mixing the leaf methanolic extract (0.1 mL) with antheron reagent (3 mL), heated at 95 °C for 15 min. The reaction mixture was then cooled in ice-chilled water, followed by incubation for 30 min at room temperature. The OD of the reaction mixture was read spectrophotometrically at 625 nm and quantified using a standard curve made from a range of pure standards (200–1000 ppm). However, NRS was calculated by the formula given below:NRS = TSS − RS

### 2.14. Determination of Malondialdehyde (MDA) Content in Leaves

The method given by Cakmak and Horst [46] was followed for MDA estimation in leaf tissue. Fresh leaf material (1 g) was well homogenized in 10 mL of 6% TCA, followed by centrifugation at 14,000× *g* at room temperature for 5 min. The supernatant obtained (0.5 mL) was reacted with 2 mL of 0.5% thiobarbituric acid (TBA) prepared in 20% TCA. After heating in water bath at 95 °C for 5 min, the mixture was then cooled in ice-chilled water. The OD of the reaction mixture was read at 532 nm and 600 nm. The following formula was used to estimate the MDA content:MDA (nmol) = Δ (A 532 nm − A 600 nm)/1.56 × 105
where the absorption coefficient for the calculation of MDA is 156 mmol^−1^ cm^−1^.

### 2.15. Determination of Leaf H_2_O_2_ Content

The method as described by Velikova et al. [47] was used for the estimation leaf H_2_O_2_ content. Leaf extract (0.1 mL) prepared in 10% TCA was mixed with 1 mL of l M KI solution, and the absorbance of resultant mixture was read at 390 nm.

### 2.16. Estimation of Stress Tolerance Indices

#### 2.16.1. Stress Susceptibility Index (SSI)

The formula given by Fisher and Maurer [48] was used for the calculation of SSI. It is the ratio of yield of each genotype under non-stressed and stressed conditions:SSI = 1 − (Ys/Yp)/SI
where Yp and Ys are the average yield of each genotype under water deficit stress and normal irrigation conditions, respectively, and SI is the stress intensity.

#### 2.16.2. Tolerance Index (TOL)

The following formula was used to calculate the tolerance index (TOL). It represents the tolerance of each genotype under water deficit stress [49]:

#### 2.16.3. Stress Tolerance Index (STI)

For the calculation of stress tolerance index (STI), the following formula was used [50]:STI = (Ys × Yp)/(Ȳp)^2^
where Ȳp is the average yield of all genotypes.

#### 2.16.4. Geometric Mean Productivity (GMP)

GMP was determined following the formula given below [51]. It is the square root of productive yield under water deficit stress and non-stress conditions, respectively:
$$GMP=\sqrt{\mathrm{Ys}}\times \mathrm{Yp}$$


#### 2.16.5. Mean Productivity (MP)

The mean productivity was the average yield of each genotype when grown under stressful conditions; it was calculated by the formula given by Fernandez [50]:MP = (Y_p_ + Y_s_)/2

#### 2.16.6. Harmonic Mean (HAM)

HAM was calculated by the formula given by Kristin et al. [52]:HAM = [2 × (Yp × Ȳs)]/(Yp + Ys)

#### 2.16.7. Yield Index (YI)

The formula of Lin et al. [53] was used for YI calculation. It is the ratio of seed yield of each genotype under stress (Ys) and average of grain yield of all genotypes under stress (Ȳs):YI = Ys/Ȳs

#### 2.16.8. Yield Stability Index (YSI)

Bouslama and Schapaugh [54] proposed the formula of YSI:YSI = Ys/Yp

#### 2.16.9. Drought Resistance Index (DI)

The formula proposed by Lan [55] was used for calculation of DI, which is given below:$$\mathrm{DI}=\frac{\left[\mathrm{Ys}\times \left(\frac{\mathrm{Ys}}{\mathrm{Yp}}\right)\right]}{\bar{Y}s}$$

Here, Yp and Ys are the seed yields of each wheat genotype when grown under water deficit stress and well-watered conditions, respectively, while Ȳp and Ȳs and are the average seed yields of all wheat genotypes when grown under water deficit stress and well-watered conditions

### 2.17. Determination of Yield Attributes

The determinations of different yield parameters, including fertile tillers/plant (FT), number of grains per spike (NOG/Spk), number of tillers per plant (NOT/plant), number of spikelets per spike (No of Spkt/Spk), grain yield/plant (GY/plant), grain weight/spike (GW/Spk), grain yield/plant (GY/plant) and hundred grain weight (100 GW), were performed manually by collecting 10 plants from each replicate, and the provided data were used for the estimation of all attributes.

### 2.18. Statistical Analysis

To study the significant differences among inbreed lines based on the studied attributes, the data were analyzed statistically using Co-STAT window version 6.3 (developed by Cohort Software, Berkley, CA, USA). The experiment was repeated in two consecutive years for the confirmation of results. The data collected for different attributes during both years were analyzed statistically but, due to similar effects, only one year of data is presented. To find out significant differences among mean values, the least significant difference (LSD) test was used at a 5% level. XLSTAT software was used for the correlations and PCA analyses for the studied attributes and the significance among the generated values against each attribute was determined using a Spearman’s correlation table.

## 3. Results

### 3.1. Growth Attributes

Significant decreases in all growth parameters, such as flag leaf area (FLA), plant leaf area (PLA), root dry weight (RDW), shoot dry weight (SDW), root fresh weight (RFW), shoot fresh weight (SFW) and plant length (PL), were recorded in all studied wheat genotypes, including parental ones under drought. However, the extent of the decreases was wheat-genotype- and growth-parameter-specific. Regarding SFW, the minimum decreases were recorded in MP8, MP2, MP1, and MP10 followed by MP5, but in relation with SDW, MP1 and MP10 were found superior to other genotypes. The decreases in RFW were comparatively lower in genotypes MP2 and MP8, followed by MP5 and MP10, but regarding the RDW, genotypes MP8 and MP1 showed a better tolerance to water stress compared to other genotypes, as well as the parental genotypes. The decrease in the FLA due to water stress was lower in genotype MP10 and MP1, followed by MP8 and MP5, in comparison with all other wheat genotypes. Furthermore, the genotypes MP1, MP2, MP5 and MP10 showed less decreases in PLA as compared with other genotypes. Though the parental genotypes also performed well in maintaining the better PLA, they were comparable in tolerance with genotypes MP1, MP5, MP8 and MP10 (Table 1).

### 3.2. Yield Attributes

The data given in Table 2 for varying yield attributes, including number of tillers per plant (NOT/plant), number of spikelets per spike (Spkt/Spk), fertile tillers per plant (FT/plant), grain weight per spike (GW/Spk), number of grains per spike (NOG/Spk), grain yield per plant (GY/plant) and 100 grain weight (100 GW), show that, under water deficit stress, these attributes were significantly negatively affected in all studied wheat genotypes, including the parental ones. However, the extent of negative impacts of water stress was genotype- and parameter-specific. Regarding FT/plant, the minimum decrease was recorded in genotypes MP3, MP9, MP7 and MP10, followed by MP2, but in case of total NOT/plant, genotypes MP3 and MP10 showed a better tolerance in maintaining the NOT/plant, followed by genotypes MP2 and MP9, as compared with the other wheat genotypes. Regarding the Spk L, no decrease was recorded in genotype MP6, and the minimum decrease was found in genotype MP10 and MP2 followed by genotypes MP1 and MP4. Regarding the number of Spkt/Spk, no change was found in MP7 and MP10. The minimum decrease was in MP8 and MP5 compared to the other studied genotypes, but the decrease in genotypes MP9 and MP3 was similar, as it was in the parental genotype Punjab. The genotypes, MP4 and MP6, were superior in this regard when compared with other genotypes. In the case of the STR Y genotype, MP8, MP9 and MP10 showed less of a decrease in response to limited water supply. The decrease in GW/Spk was the minimum in genotypes MP10 and MP8, followed by MP5, MP3 and MP1. Genotypes MP10 and MP8 performed better in comparison to the parental genotypes regarding GW/Spk under water deficit stress. Regarding NOG/Spk, genotypes MP8 and MP10 showed better tolerance as compared with other genotypes. Furthermore, the genotypes MP8 and MP10, followed by MP5, were found to be the superior to the other genotypes in the case of better maintenance of 100 GW under water deficit conditions. Regarding the GY/plant, the minimum decrease due to water deficit stress was found in genotype MP5, followed by in genotypes MP8 and MP10, as compared with the other genotypes and they were also found superior to the parental genotypes. Overall, the genotypes MP5, MP8 and MP10 showed a better performance under water deficit stress when were compared with the other genotypes, including the parental genotypes regarding all studied yield attributes.

### 3.3. Photosynthetic Attributes and Water Relations

A reduced water supply significantly affected the leaf total chlorophyll (T. Chl.), chlorophyll b (Chl. *b*), chlorophyll *a* (Chl. *a*), and chlorophyll *a*/chlorophyll *b* (Chl. *a/b*) of all the studied wheat genotypes. Leaf Chl. *a* content increased significantly in genotypes MP2, MP5 MP8 and Millat, but decreased significantly in all other genotypes, including in genotype Punjab. The maximum increase in Chl. *a* under water deficit stress was found in genotype MP8, while the minimum decrease was found in genotype MP1 followed by in MP9 and MP10. An increase in Chl. *b* due to water deficit stress was found in genotypes MP1, MP2, MP8 Millat and Punjab, but a decrease was found in MP4, MP5, MP6 and MP7 and MP10. The maximum increase in Chl. *b* under water deficit stress was found in genotype MP8, while the minimum decrease was in genotypes MP5 and MP10. Regarding Chl. *a/b*, an increase was recorded in genotypes MP4, MP6, and MP7 and no significant increase or decrease was found in other genotypes. However, the genotypes MP4, MP5 and MP6 were superior to the other genotypes when grown under a limited water supply. In the case of leaf T. Chl. content, an increase was found in genotype MP2, MP8, Millat and Punjab, and a significant reduction was recorded in other wheat genotypes, except in those of MP1 and MP5, in which no significant increasing or decreasing effect of water deficit stress was found. The maximum increase in leaf T. Chl. was recorded in genotype Millat followed by MP8, while the minimum decrease was found in genotype MP10 followed by MP9 (Table 3).

Water stress also significantly increased the leaf carotenoid (Car) content of all studied wheat genotypes, except for MP2, MP4, MP7 and Millat where the increase was non-significant. The maximum increase in leaf Car under limited water supply was in genotype MP6 followed by MP5. However, leaf T. Chl./Car decreased significantly due to water deficit stress in all genotypes, except for genotypes MP2 and MP3 where no significant increase or decrease was recorded. The minimum decrease in leaf T. Chl./Car was found in genotype MP10 followed by MP1, MP8 and MP9, as compared with other genotypes. Moreover, genotypes MP10, MP8 and MP1 maintained better T. Chl./Car than other genotypes (Table 3).

A reduced water supply also significantly decreased the leaf relative water content (LRWC) of all studied wheat genotypes including the parental ones. However, the decreasing effect was wheat-genotype-specific. A comparatively lower decrease in LRWC was recorded in genotypes MP1, MP5, MP8 and MP 10, as compared to other genotypes. The maximum decrease in LRWC was found in genotypes MP2, MP4 and MP6, as compared with other genotypes (Table 3).

### 3.4. Leaf Hydrogen Peroxide (H_2_O_2_)

Leaf hydrogen peroxide (H_2_O_2_) levels significantly increased in all studied wheat genotypes under limited water supply. However, the extent of increase was wheat-genotype-specific. The minimum increase was recorded in genotypes MP10 followed by MP5, MP6 and MP1 in comparison with other wheat genotypes. While the genotypes Millat, Punjab and MP3, respectively showed the least stress tolerance in this regard with more increase in leaf H_2_O_2_ content as compared with all other genotypes under limited water supply. However, the other genotypes were intermediate in H_2_O_2_ accumulation (Figure 1A).

### 3.5. Leaf Enzymatic Antioxidants

The leaf superoxide dismutase (SOD) and peroxidase (POD) activities of all studied wheat genotypes were significantly affected when grown under water deficit stress. However, the increasing and decreasing effect was wheat-genotype-specific. Leaf SOD activity increased in all studied wheat genotypes due to water deficit stress, except in genotypes MP2, MP3 MP4, and MP6 where it significantly decreased. The maximum significant increase in leaf SOD activity was found in genotypes Millat, Punjab, and MP8, followed by genotypes MP1, MP5, MP9, MP10 and MP7, but the minimum decrease in SOD activity due to reduced irrigation was found in genotype MP4, followed by in MP6 (Figure 1B). Regarding the leaf POD activity, a significant reduction was recorded in genotypes MP5, MP8, Punjab, Millat, MP2, and MP6, but this decrease was significant in genotypes MP4, MP3, MP7, MP1, MP9 and MP10. The maximum increase in leaf POD activity was found in MP5 followed by MP8. The opposite was true for MP4, MP3, MP7, and MP1, but the smallest decrease was recorded in genotype MP10 (Figure 1E).

Drought stress also significantly affected the leaf catalase (CAT) and ascorbate peroxidase (APX) activities in all studied wheat genotypes but the increasing or decreasing effect was wheat-genotype-specific. Regarding the leaf CAT activity, it was found that its activity increased significantly in genotypes MP8, Millat, Punjab, MP5, MP6, MP10, MP2 and MP1, but decreased significantly in genotypes MP4, MP7, MP9 and MP3. The maximum increase in leaf CAT activity was found in genotype MP8 followed by genotype MP5, MP6 and MP10 (Figure 1C), while the minimum decrease was found in genotype MP3 as compared with other genotypes. A significant increase in leaf APX activity was recorded in wheat genotypes when grown under water deficit stress, except in genotypes MP4, MP3, MP10 and MP6 where it decreased significantly. The maximum increase in leaf APX activity was recorded in genotypes MP8 and MP5 and the maximum decrease in leaf APX activity was found in genotype MP4 (Figure 1D).

### 3.6. Leaf Malondialdehyde (MDA) and Leaf Relative Membrane Permeability (LRMP)

A significant increase in leaf malondialdehyde (MDA) content was recorded in all studied wheat genotypes when grown under water deficit stress. The maximum relative increase in MDA under water deficit conditions was recorded in genotypes Punjab, and Millat, while the minimum was recorded in the genotypes MP10 and MP7. However, the other genotypes showed comparatively the intermediate response of accumulating the MDA (Figure 1F).

Data given in Figure 1 show that a limited water supply significantly increased the leaf relative membrane (LRMP) in all studied wheat genotypes, including the parental genotypes. However, the extent of increase was wheat-genotype-specific. A relatively lower increase in LRMP was recorded in genotypes MP1, MP2, and MP5, followed by genotype MP10 as compared with other genotypes, while the genotype MP9 was the inferior genotype in this regard (Figure 1G).

### 3.7. Leaf Flavonoid Contents, Total Phenolics Content (TPC) and Leaf Ascorbic Acid (AsA) Content

Leaf ascorbic acid (AsA) content increased significantly in all studied wheat genotypes when grown under a limited water supply. However, this increase was wheat-genotype-specific. The maximum increase in leaf AsA content was found in the genotype Millat, followed by genotypes Punjab, MP10, MP9, MP8, MP3, MP2 and MP5, respectively. However, the lowest increases in leaf AsA were found in genotypes MP4, MP6 and MP1, respectively (Figure 2).

Leaf TPC and flavonoids contents were also significantly affected due to the imposition of water deficit stress in all the studied wheat genotypes. However, the increasing or decreasing effect was wheat-genotype-specific. Regarding the leaf TPC, it increased significantly in all studied wheat genotypes, except the genotypes MP2, MP3, MP10 and Punjab where it decreased significantly. The higher increase in leaf TPC was found in genotypes MP8, MP4, MP9, MP5 and MP7, followed by MP1 and Punjab. However, the maximum decrease was in genotypes MP2 and MP3, followed by in genotype MP10 and Punjab, respectively, yet no significant effect of water deficit stress was recorded on TPC in genotype MP6. Moreover, the leaf flavonoids content significantly increased in all studied wheat genotypes when grown under limited water supply, except in genotypes MP4 and MP7 where it decreased significantly; however, no significant effect was recorded in genotypes MP6 and Punjab. The most increments in leaf flavonoids under limited water supply were found in genotypes MP10, MP8 and Millat, followed by in genotypes MP9, MP1 and MP5, comparative to other genotypes (Figure 2).

### 3.8. Total Soluble Sugars (TSS), Non-Reducing Sugars (NRS), Reducing Sugars (RS), Leaf-Free Amino Acids (FAA), Total Soluble Proteins (TSP) and Anthocyanin Contents

The imposition of drought stress significantly increased the total soluble sugar (TSS), free amino acids (FAA), non-reducing sugars (NRS) and reducing sugars (RS) contents of all wheat genotypes except to that of genotype MP7 in case of FAA; MP4 in case of TSS and NRS, MP3, MP4 and MP7 in case of RS, respectively, where a significant decrease was recorded due to water deficit stress. However, the extent of increases in FAA, RS, NRS and TSS was genotype-specific. Regarding NRS and TSS genotypes, MP10, MP6, MP9, MP8, MP5 and MP2, including Millat and Punjab, respectively, showed more increases when compared with other genotypes under limited water supply. The higher increases were recorded in genotypes MP8, MP9 and MP10, and the minimum increase was recorded in genotype MP5. Regarding the RS, the higher increment was found in genotypes Millat, MP8, MP5 and MP10, while the minimum increase was found in genotype MP2. Moreover, the maximum increase in FAA was found in genotypes Millat, Punjab, MP8, MP1 followed by in MP2, and the minimum increase was found in genotype MP5 (Figure 3).

Leaf anthocyanin and the total soluble protein (TSP) contents of all wheat genotypes were also significantly affected when grown in limited water supply but the decreasing or increasing effect was wheat-genotype-specific. Moreover, the extent of increase or decrease in leaf TSP and anthocyanin was genotype-specific. Regarding TSP, a significant increase was recorded in genotypes Millat, Punjab, MP8, and MP5, followed by in MP3 and MP2 under limited water supply while genotypes MP4, MP9, MP6 and MP7 showed a significant decrease under limited water supply. The higher increases were recorded in genotypes MP8 and MP5, followed by MP3, while the maximum decrease was recorded in genotype MP4, in comparison to other genotypes. In case of leaf anthocyanin content, a significant increase was recorded in genotypes MP8, Millat, Punjab, MP10, MP9, MP5, and MP2, while a significant decrease was recorded in genotypes MP7, MP6, MP3 and MP1. The higher increment in leaf anthocyanin was found in genotype MP8 followed by genotypes Millat, Punjab, MP5, MP9 and MP10, as compared to the other genotypes, while the maximum decrease was recorded in genotypes MP1, followed by genotype MP3 when under water deficit stress (Figure 3).

### 3.9. Stress Tolerance Indices (STI)

Table 4 depicts the data regarding the different stress tolerance indices. Based on these stress tolerance indices, a selection of drought-sensitive and drought-tolerant genotypes was made in combination with different physio-biochemical attributes. Regarding STI, GMP and HAM, genotypes MP1, MP3, MP5, MP8 and MP10 were found to be more tolerant. However, genotypes MP1, MP5, MP8, MP7 and MP10 performed better in terms of MP. In the case of YI, genotypes MP1, MP5, MP3 and MP8 were superior to the other genotypes, while in case of YSI, genotypes MP5, MP8 and MP10 performed better. Moreover, in case of DI, genotypes MP1, MP5, MP8 and MP10 were highly tolerant and even performed better than parental genotypes. Based on SSI genotypes MP6, MP2 and MP9 were found to be the most sensitive than other genotypes, while regarding TOL, genotypes MP1, MP2, MP3 and MP6 showed more sensitivity.

### 3.10. Correlations Studies

Correlation data for varying yield and growth parameters in relation with biochemical and physiological attributes are presented in Table 5. The number of grains/Spk, GW/Spk, 100 GW and GY/plant have strong positive correlations with RFW, RDW, SFW, and SDW. RFW, RDW, SFW, and SDW have a non-significant correlation with photosynthetic attributes, such as T. Chl. Chl. *b* and Chl. *a*, but have a strong negative correlation was with leaf Car content. However, leaf T. Chl./Car was found to be strongly positively correlated with different morphological and growth parameters. The number of grains/Spk (−0.652 and −0.797), GW/Spk (−0.673 and −0.818), 100 GW (−0.502 and −0.681), and GY/plant (−0.533 and −0.507) negatively correlated with leaf H_2_O_2_ and MDA, respectively. Growth attributes were also found to be negatively correlated with leaf MDA and H_2_O_2_ contents.

The PCA analysis of different studied attributes presented in Figure 4 shows a close relationship of different yield (NOG/Spk, GW/Spk, 100 GW, GY/plant) and growth attributes (FLA, PLA, SFW, RFW, SDW, RDW and PL). The activities of antioxidant enzymes such as SOD and APX and non-enzymatic antioxidants (TPC, Car, flavonoids and anthocyanin) were closely positively related but strongly negatively related with yield and growth attributes. The accumulation of MDA and H_2_O_2_ was closely related and had a close relation with FAA, AsA, RS, TSS, NRS and LRMP sensitivity.

## 4. Discussion

Breeding to enhance the yield of crops with abruptly changing environmental conditions is the prime focus for breeders/researchers to meet the increasing demand for food caused by a rapidly increasing world population. The behavior of plants under such changing environmental conditions is a challenge to the production of consistent, high-yielding crop varieties, especially regarding cereals, which are considered to be the most important crops needed to fulfill the world food demand. These crops demonstrate the abrupt and unexpected changing behavior of the stressful environment [12,26]. In crop plants, the selection of high-yielding crop varieties by agronomists for growth and better production under such abruptly changing environmental conditions is mainly based on agronomic traits [56]. However, very little importance was given to physiological and biochemical mechanisms; their activity appears in phenotypic characteristics, including agronomic traits, due to their firm genetic attachment, especially under changing environmental conditions [19,57,58]. The morphological adaptations under water deficit stress include denser, deeper, and extended roots; reduced shoot growth to maintain a high root/shoot ratio; and delayed senescence, while the high content of chlorophyll, lower osmotic potential, and waxy leaf coverings indicate biochemical adjustments. Different physiological adaptations include the balanced photosynthetic activity and production of fatal metabolites [59]. Aside from these, other changes that occur under water deficit stress include damages to photosynthetic apparatus and components, disruptive effects on the enzymes of the Calvin cycle, leading to crop yield reductions due to oxidative stress, the latter of which is also considered to be a key factor responsible for reduced photosynthesis, leading to yield losses [60].

In crop plants, the effects of adverse environmental conditions, such as those of water stress, are mainly measured by focusing on reductions in growth and yield that are strongly and primarily linked with plant–water relations and should be considered of prime importance in selection breeding. Other physiological mechanisms, such as plant photosynthetic efficiency, membrane integrity, antioxidative defense mechanisms, and nutrient acquisition are also strongly dependent on the adjustment of plant–water relations. However, in the development of stress-tolerant genotypes through breeding techniques, there is very little focus on the interactive roles to such mechanisms, along with other physiological mechanisms [57]. The crop genotypes that show tolerance against the adverse environmental conditions have a better capacity of cellular osmotic adjustment for maintaining plant–water relations, as well as reduced lipid peroxidation by maintaining the antioxidative defense mechanism. Therefore, the tolerance in crop plants against stressful environments is the combined effect of genotypic expression and physiological alterations that lead to phenotypic modulations with better agronomic performance [19,61].

Plant biomass production under water deficit conditions is linked with the better adjustment of water relations to maintain physiological and biochemical processes [19,28,62]. It is known that, to survive under water deficit stress, wheat experiences phenotypic changes and altered dry matter partitioning, e.g., a reduced biomass production and leaf area, a smaller decrease in root growth and reduced injury [63] that shows the adoptability for balancing the water status of plant tissues [64,65]. Moreover, the leaf longevity, shrunken leaf size and decreased number of leaves are also the adaptabilities for survival and better production under water deficit stress [66], which are genetically controlled and genotype-specific [67,68,69]. In the present findings, the significant negative impacts of limited water supply were found for different morphological traits, biomass production and yield-related attributes of all the studied wheat genotypes. However, the impacts were genotype-specific, and the extent of decrease shows the relative tolerance of wheat genotypes to water deficit stress. Genotypes MP1, MP8, MP5, and MP10 performed better in relation to the studied growth attributes under stressful environmental conditions. Significant reductions were recorded in FLA, PLA, SFW, SDW, RFW, RDW and PL in all 12 of the studied genotypes, but a significantly lower reduction was recorded in genotypes MP1, MP8, MP5 and MP10, as compared with other genotypes that showed their better ability to tolerate water deficit stress. Moreover, wheat genotypes MP1, MP8, MP5, MP10, followed by MP3, also performed better regarding the different yield attributes, such as the number of Spkt/Spk, NOG/Spk, GW/Spk, 100 GW and GY/plant, and showed a lower decrease in these attributes as compared with other genotypes. It is well known that genotypes which perform better in maintaining the biomass under water stress show better results in producing better seed yield [70,71]. Moreover, these wheat genotypes were also found better when studied for different STIs in relation with yield attributes, as has been presented in Table 4. However, very little is reported regarding the use of STI for the selection of drought-tolerant wheat genotypes in combination with the physio-biochemical mechanisms. It was reported that genotypes with higher values of STI are considered to be stress-tolerant genotypes [72]. In view of the reports made by Nouraein et al. [73], the STI were found to be better for the selection of stress-tolerant crop genotypes for growth under water deficit stress. In the present study, genotypes MP1, MP3, MP5, MP8 and MP10 were also found to be categorized as stress-tolerant genotypes based on STI in combination with the physio-biochemical mechanisms.

The maintenance of better growth and yield in genotypes MP1, MP5, MP10 and MP8 under water deficit stress is also well-correlated with the maintenance of better leaf photosynthetic pigments. An increase was recorded in Chl. *a* for genotypes MP5 and MP8, in Chl. *b* for genotypes MP1 and MP8, but a lower decrease was recorded in Chl. *a* for genotypes MP1 and MP10 and Chl. *b* for genotype MP5 and MP10. Similarly, regarding T. Chl., an increase was found in genotype MP8, a lower decrease in MP5 and MP3, and no effect was recorded in genotype MP1. Therefore, the genotypes, MP1, MP5, MP10 and MP8, tend to maintain the leaf photosynthetic pigments under water stress, which shows their relative stress tolerance ability. Leaf chlorophyll is also considered as one of the most commonly used metrics for assessing the severity of drought stress [74,75]. It is stated that, in crop plants, the green leaf area corresponds well with their photosynthetic capacity and better productivity, as well as phenological and physiological status, especially under water deficit stress [76,77,78] and the leaf photosynthetic efficiency is the function of light capturing ability and leaf stomatal regulation. The first case is dependent on the better photosynthetic pigments and the latter case is dependent on plant water status. Studies revealed that leaf chlorophyll contents correlated well with yield-related attributes and can detect the health status of the plant under water deficit stress [79,80,81]. Drought-tolerant varieties maintain better chlorophyll contents and vice versa [82]. The maintenance of proper chlorophyll contents under water deficit stress is a purely under genetic control in crop plants [83]. It was reported that the content of leaf chlorophyll also affects yield traits, such as grain yield and number of grains [84,85], which signify the importance of leaf chlorophyll contents, especially under water deficit stress [86,87,88]. In the present findings the better performance of genotypes MP1, MP5, MP10, and MP8, in relation with the growth and yield attributes, is correlated well with their better maintenance of photosynthetic pigments. This means that there is a greater light-capturing ability with the maintenance of better photosynthetic pigments [89,90,91]. The reduction in photosynthetic pigments in other genotypes can be correlated with the increased oxidative stress and lipid peroxidation [78,92], which are typical symptoms of chlorophyll reduction under water deficit stress. It was found that changes in photosynthetic pigments and components, along with the damages to photosynthetic apparatus, resulted in crop yield losses [60] and the similar was found in the present findings.

The primary impact of limited water supply is on the water relations of plants that determine the ability of a plant to tolerate water-deficit conditions regarding growth, yield and photosynthetic efficiency. In screening, the drought-tolerant genotypes LRWC are considered the most important parameter. This indicates the extent of membrane stability and balance between evapotranspiration and water supply. Drought-tolerant genotypes retain more cellular water content when compared with drought-sensitive ones [19]. The most effective genotypic variation, under water deficit conditions and regarding the better LRWC, is having the potential to uptake more water from the soil or the plant’s ability to maintain a lower osmotic potential for the maintenance of the tissue turgor, leading to better physiological activities for a better seed yield [19,93]. The genotypes MP1, MP5, MP8 and MP10 maintained their better LRWC in the present study, showing their better ability to tolerate water deficit conditions with better growth, photosynthetic pigments and yield, as compared with other genotypes.

It is known that tolerant genotypes acquire their better LRWC by solute accumulation and metabolites, known as cellular osmoregulation [94,95]. The maintenance of better LRWC under water stress is dependent on better cellular osmotic adjustment due to the accumulation of inorganic nutrients, metabolites, amino acids, and less lipid per oxidation due to reduced leaky membranes [94,95,96,97]. The genotypes MP8, MP1, and MP5 accumulated more FAA; genotypes MP8, MP5, and MP10 accumulated more RS; genotypes MP8, MP10 and MP5, including the parental genotypes, accumulated more NRS and TSS; and genotypes MP8, MP5, and MP3 accumulated more TSP. This better osmotic adjustment in genotypes MP8, MP10, and MP3 is positively correlated with their better LRWC, which shows their greater ability to tolerate water deficit conditions by maintaining better water contents.

In view of the cellular osmotic adjustment under water deficit stress the accumulation of sugars is also considered as an organic osmolyte and has been reported in many plant species [98,99,100]. While working on wheat, it was found by Chen et al. [101] that sugar accumulation is considered to be a physiological marker for comparing different wheat genotypes because a high sugar accumulation maintains the protein structure, enzyme activities, osmotic adjustments, signal transduction, cell structure stability functions and cellular redox balance [102]. Cellular sugar accumulation also plays a crucial role in coping with moisture deficiency by maintaining LRWC in a genotypic association manner [84], as was also found in the present study. Moreover, in the present findings, the maintenance of better chlorophyll pigments can be associated with the higher accumulation of sugars, which also act as osmoprotectant and maintain better cellular activities. This helps to tolerate drought stress with a better cell membrane in tolerant wheat genotypes when grown with a limited water supply [101]. In the present study, LRMP was also lower in genotypes, MP10, MP5 and MP1, as compared with other genotypes that maintained a better cell turgidity, necessary for better growth. In these genotypes, it was also linked with better biomass production, photosynthetic pigments and seed-yield-related attributes. Similar trends were also recorded in previous studies, where the maintenance of better water relations in wheat genotypes was associated with the better accumulation of free amino acids, proline, and total soluble protein, along with less membrane damages, leading to better biomass production and seed yield [95].

The better performance of the plants under water deficit stress is also found to be associated with a better performance of the antioxidative defense system, maintaining the integrity of membranes through a better scavenging mechanism for overly produced ROS [21,103]. This means that the cultivars with better antioxidative defense mechanisms not only show reduced damages to chloroplastic membranes in maintaining electron transport, but also less leaky cellular membranes that maintain a better cell turgidity, essential for better growth and water relations [104]. Damages to cellular membranes due to overly produced ROS are studied in terms of MDA accumulation, and the extent of MDA production represents the severity of oxidative stress along with the potential of antioxidative defense [105,106]. The oxidative stress results in reductions in growth and yield through several disturbances in normal metabolic activities. Plants evolved a well-developed antioxidative defense system to counteract the damaging impacts of oxidative stress. However, the antioxidative defense system is plant-species- and even cultivar-specific [107,108]. It includes the enzymatic activities of POD, SOD, APX and CAT, and the accumulation of non-enzymatic antioxidants such as AsA, phenolics, carotenoids, flavonoids, and anthocyanin. Both of these types of antioxidantive mechanisms work together in a supportive way [19,106,109,110,111]. However, this antioxidative mechanism is complicated and genetically controlled [111,112,113]. In wheat, this antioxidative mechanism is more complicated under adverse environmental conditions due to its complex genome, where most of its traits are multigenic [17,114,115].

In present study, genotypes MP1, MP5, MP8 and MP10, including parental genotypes performed better regarding the activities of antioxidant enzymes such as CAT, POD, SOD and APX, as well as the accumulation of non-enzymatic antioxidants, including AsA, phenolics, flavonoids, and anthocyanin. The results of the present study can be correlated with previous studies, where the activities of different antioxidants were increased significantly in different wheat genotypes under stressful environmental conditions but the increase was genotype-specific [107,108,116,117]. Upadhyay et al. [118] reported that improved activities of different enzymatic antioxidants, in combination with non-enzymatic antioxidants, helped to alleviate the damages of ROS to the cellular membranes in four wheat genotypes. Similar results to those found in the present study were reported by Chakraborty and Pradhan [116] in different wheat cultivars, where increased activities of different antioxidative enzymes and the content of non-enzymatic antioxidants in response to oxidative stress resulted in less leaky membranes due to overly produced ROS; the efficiency of the oxidative stress tolerance mechanism is measured based on MDA accumulation and the levels of ROS such as H_2_O_2_. The genotypes with better antioxidative defense mechanisms in terms of decreased MDA accumulation were considered tolerant. Such genotypes have better enzymatic and non-enzymatic antioxidative defense mechanisms [106,110]. The overly produced ROS damages the cellular membranes, which become leaky and result in a decrease in cell turgor potential, leading to disturbed cellular water relations. Damages to thyllakoid membranes because of overly produced ROS under limited water supply is also well known, causing a reduction in leaf photosynthetic pigments and its components. These damages to the photosynthetic membranes lead to their reduced functioning [60].

In the present study, low levels of H_2_O_2_ were found in genotypes MP10, MP8, MP5 MP1 and MP3 with a reduced lipid peroxidation that is linked with their better antioxidative defenses regarding the increased activities of antioxidant enzymes (CAT, POD, SOD and APX) and a higher accumulation of non-enzymatic antioxidants (AsA, anthocyanin, carotenoids, phenolics and flavonoids). It can also be correlated well with better photosynthetic pigments in these genotypes due to less degradation of lipids per oxidation of chloroplastic membranes, resulting in the maintaining of a better chlorophyll content. Moreover, the H_2_O_2_ contents in these genotypes were lower, confirming their better antioxidation capacity to capture the overly produced ROS that led to a reduction in lipids per oxidation and resulting in better plant growth and yield. Moreover, the reduced LRMP in parallel with better LRWC, further conferred their excellent antioxidative defense mechanisms. It was found that less H_2_O_2_ content and MDA accumulation in different wheat varieties was dependent on a better antioxidative defense potential. This might be due to the maintenance of better cellular water content that resulted in better growth [119].

## 5. Conclusions

Overall, in view of the findings of this study, it can be concluded that the better drought tolerance of wheat inbred lines MP1, MP5, MP8, MP10 (and MP3 for better yield and growth attributes) is associated with the better adjustment of cellular water relations by osmotic adjustment, reduced lipid peroxidation due to better antioxidative defense mechanisms, and the maintenance of better photosynthetic pigments. The lines MP1, MP5, MP8 and MP10 can be used for better yield under a limited water supply, as well as a genetic source for further breeding programs to induce drought tolerance in wheat genotypes, which will help to solve the problem of increasingly world food demand to some extent.

## Figures and Tables

**Figure 1 plants-11-00466-f001:**
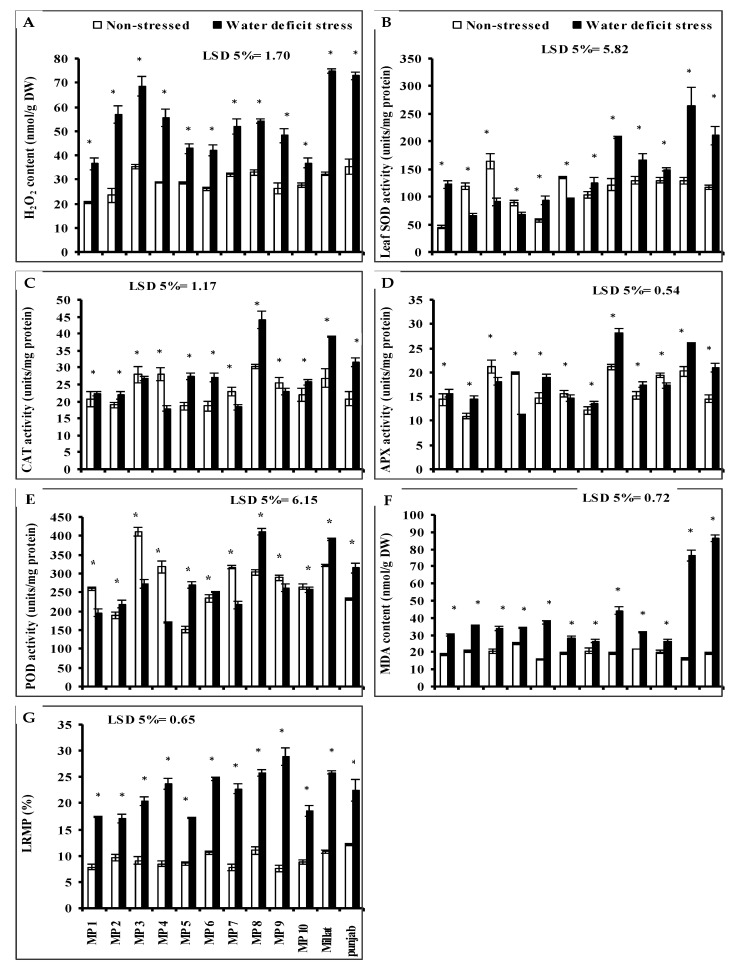
(**A**) Leaf hydrogen peroxide (H_2_O_2_), (**B**) superoxide dismutase (SOD), (**C**) activities of catalase (CAT), (**D**) ascorbate peroxidase (APX), (**E**) peroxidase (POD), (**F**) content of malondialdehyde (MDA) and (**G**) leaf relative membrane permeability (LRMP) of the newly developed inbreed wheat genotypes under limited water supply and normal irrigation (Mean ± SE; *n* = 4). (Bars against a wheat genotype with * shows significance effect of water deficit stress).

**Figure 2 plants-11-00466-f002:**
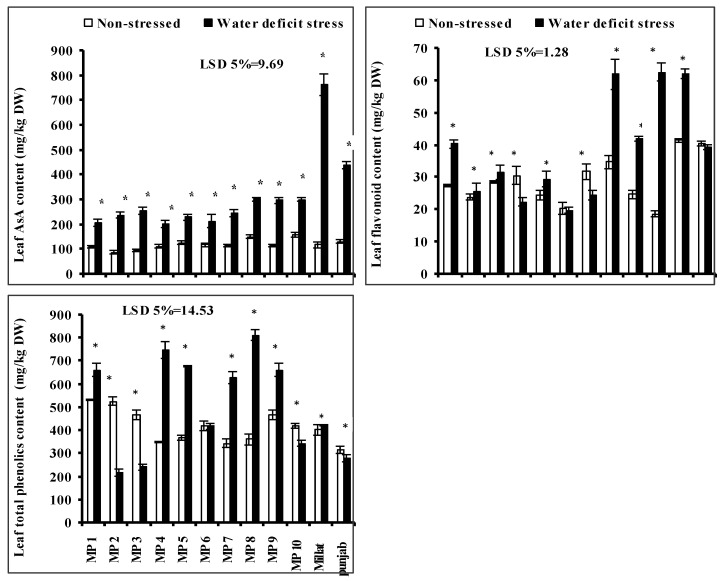
Leaf total phenolic contents (TPC), flavonoids and ascorbic acid (AsA), of newly developed inbreed wheat genotypes when grown under limited water supply and normal irrigation (Mean ± SE; *n* = 4). (Bars against a wheat genotype with * shows significance effect of water deficit stress).

**Figure 3 plants-11-00466-f003:**
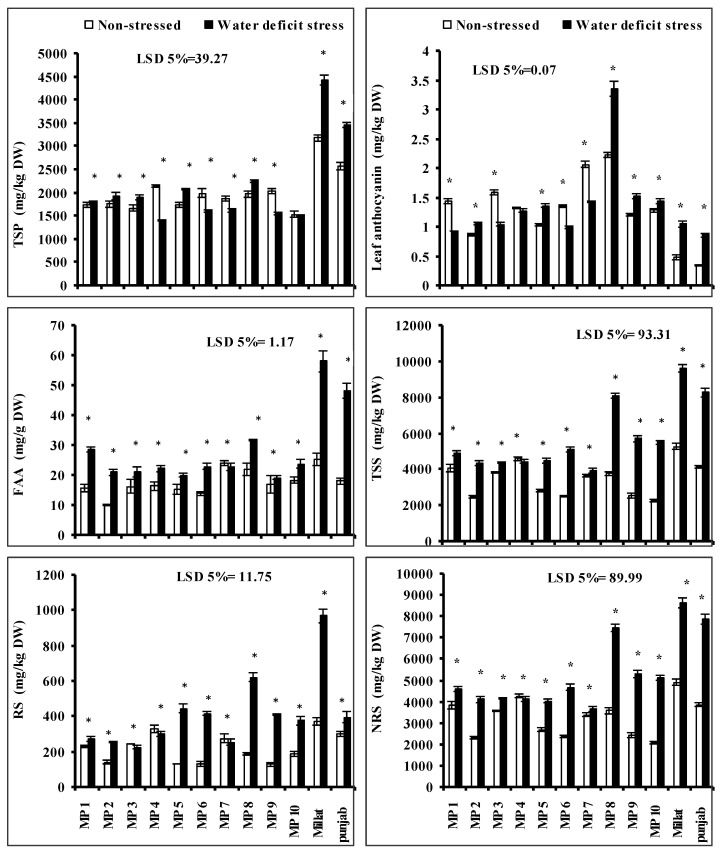
Total soluble sugars (TSS), non-reducing sugars (NRS), reducing sugars (RS), leaf-free amino acids (FAA), total soluble proteins (TSP), and anthocyanin contents of newly developed inbreed wheat genotypes under limited water supply and normal irrigation (Mean ± SE; *n* = 4). (Bars against a wheat genotype with * shows significance effect of water deficit stress.).

**Figure 4 plants-11-00466-f004:**
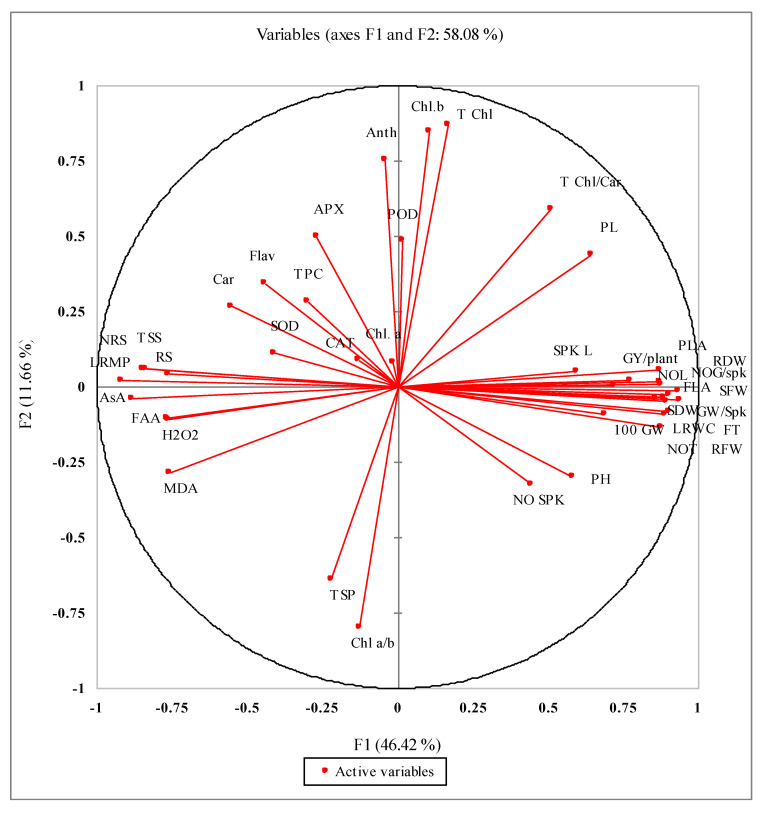
Principal component analyses (PCA) for different growth, biochemical and physiological attributes of newly developed wheat genotypes under limited water supply and normal irrigation.

**Table 1 plants-11-00466-t001:** Morphological attributes of newly developed wheat genotypes grown under water deficit stress and normal irrigation (Mean ± SE; *n* = 3).

	FLA (cm^2^)		PLA (cm^2^)		SFW (g/plant)		RFW (g/plant)	
Genotypes	Normal Irrigation	Water Deficit Stress	Normal Irrigation	Water Deficit Stress	Normal Irrigation	Water Deficit Stress	Normal Irrigation	Water Deficit Stress
MP 1	* 41.4 ± 3.8 ^cd^	25.7 ± 1.3 ^bc^	* 3807 ± 422 ^c^	2271 ± 128 ^bc^	* 224 ± 8.0 ^c^	132 ± 7.1 ^a^	* 24.4 ± 1.8 ^f^	15.0 ± 0.7 ^a^
MP 2	* 39.5 ± 1.2 ^d^	20.5 ± 3.7 ^f^	* 2522 ± 258 ^e^	1511 ± 278 ^cd^	* 185 ± 9.3 ^e^	113 ± 6.8 ^c^	* 19.9 ± 1.6 ^j^	13.8 ± 1.4 ^b^
MP 3	* 55.7 ± 2.1 ^a^	22.7 ± 2.1 ^de^	* 6456 ± 488 ^a^	1336 ± 53 ^d^	* 309 ± 9.3 ^a^	124 ± 6.0 ^b^	* 40.7 ± 1.9 ^a^	14.6 ± 1.3 ^ba^
MP 4	* 45.2 ± 1.8 ^b^	20.9 ± 0.5 ^f^	* 4561 ± 525 ^b^	1053 ± 104 ^d^	* 309 ± 6.8 ^a^	85 ± 4.2 ^e^	* 32.4 ± 2.8 ^c^	14.4 ± 0.8 ^ba^
MP 5	* 45.1 ± 1.9 ^b^	26.5 ± 0.7 ^b^	* 3166 ± 271 ^d^	1867 ± 47 ^c^	* 222 ± 8.5 ^c^	126 ± 3.7 ^b^	* 23.9 ± 1.5 ^g^	14.9 ± 0.8 ^ba^
MP 6	* 42.6 ± 1.4 ^c^	16.3 ± 1.2 ^g^	* 4319 ± 278 ^b^	9556 ± 72 ^a^	* 261 ± 3.2 ^b^	62 ± 3.8 ^g^	* 33.9 ± 2.0 ^b^	10.2 ± 0.2 ^c^
MP 7	* 46.5 ± 1.0 ^b^	21.5 ± 1.7 ^ef^	* 4366 ± 290 ^b^	1005 ± 57 ^d^	* 261 ± 9.4 ^b^	74 ± 4.1 ^f^	* 28.5 ± 1.4 ^d^	8.3 ± 0.9 ^d^
MP 8	* 56.3 ± 1.4 ^a^	33.5 ± 0.6 ^a^	* 4346 ± 349 ^b^	2398 ± 51 ^b^	* 188 ± 5.4 ^e^	117 ± 5.5 ^c^	* 20.4 ± 1.3 ^ij^	13.5 ± 0.9 ^b^
MP 9	* 40.7 ± 1.3 ^d^	20.7 ± 1.2 ^f^	* 3148 ± 102 ^d^	1151 ± 76 ^d^	* 226 ± 8.3 ^c^	101 ± 6.8 ^d^	* 27.3 ± 1.8 ^e^	12.8 ± 1.0 ^b^
MP 10	* 34.3 ± 1.6 ^f^	21.4 ± 0.7 ^ef^	* 2484 ± 190 ^e^	1450 ± 143 ^cd^	* 199 ± 9.0 ^d^	117 ± 6.9 ^c^	* 21.1 ± 1.9 ^i^	13.1 ± 0.2 ^b^
Millat	* 36.7 ± 1.3 ^e^	21.4 ± 2.0 ^ef^	* 1992 ± 227 ^f^	1038 ± 156 ^d^	* 170 ± 9.9 ^f^	104 ± 6.1 ^d^	* 20.7 ± 1.2 ^ij^	9.7 ± 0.3 ^c^
Punjab	* 42.5 ± 1.1 ^c^	24.2 ± 0.5 ^cd^	* 2773 ± 234 ^de^	1636 ± 110 ^cd^	* 195 ± 8.8 ^d^	113 ± 7.0 ^c^	*22.5 ± 1.5 ^h^	12.7 ± 1.2 ^b^
LSD 5%	1.78		494		6.64		1.14	
	SDW (g/plant)		RDW (g/plant)		PL (cm)			
Genotypes	Normal irrigation	Water deficit stress	Normal irrigation	Water deficit stress	Normal irrigation	Water deficit stress		
MP 1	* 78 ± 2.6 ^a^	48 ± 0.6 ^a^	* 18.3 ± 1.1 ^c^	11.7 ± 0.2 ^gh^	* 112 ± 0.76 ^a^	107 ± 1.02 ^a^		
MP 2	* 65 ± 3.9 ^d^	37 ± 1.0 ^c^	* 16.7 ± 0.3 ^d^	8.7 ± 0.4 ^bc^	* 107 ± 1.52 ^c^	102 ± 0.76 ^c^		
MP 3	* 77 ± 5.0 ^ba^	36 ± 1.6 ^c^	* 19.4 ± 2.5 ^b^	9.1 ± 0.4 ^bc^	* 109 ± 0.51 ^b^	102 ± 1.52 ^c^		
MP 4	* 77 ± 1.7 ^ba^	30 ± 3.3 ^de^	* 21.3 ± 1.8 ^a^	9.4 ± 0.6 ^b^	* 107 ± 0.76 ^c^	99 ± 0.76 ^d^		
MP 5	* 70 ± 4.1 ^c^	40 ± 0.9 ^b^	* 15.4 ± 1.7 ^e^	9.0 ± 0.6 ^bc^	* 107 ± 1.78 ^c^	99 ± 0.76 ^d^		
MP 6	* 75 ± 5.0 ^b^	23 ± 0.9 ^f^	* 14.8 ± 1.3 ^ef^	6.2 ± 0.4 ^d^	* 107 ± 1.52 ^c^	97 ± 1.78 ^e^		
MP 7	* 76 ± 3.2 ^ba^	28 ± 2.7 ^e^	* 13.7 ± 0.9 ^f^	5.7 ± 0.4 ^d^	* 104 ± 1.52 ^d^	99 ± 1.52 ^d^		
MP 8	* 51 ± 2.7 ^f^	26 ± 0.4 ^ef^	* 12.0 ± 0.5 ^g^	8.2 ± 0.4 ^c^	* 112 ± 0.76 ^a^	104 ± 1.52 ^b^		
MP 9	* 65 ± 3.6 ^d^	32 ± 0.6 ^d^	* 14.4 ± 1.9 ^f^	8.5 ± 0.6 ^c^	* 112 ± 0.76 ^a^	102 ± 1.78 ^c^		
MP 10	* 59 ± 1.8 ^e^	39 ± 0.3 ^bc^	* 11.2 ± 0.4 ^h^	5.7 ± 0.3 ^d^	* 112 ± 0.76 ^a^	107 ± 1.52 ^a^		
Millat	* 47 ± 2.9 ^g^	19 ± 0.2 ^g^	* 10.7 ± 0.7 ^h^	4.8 ± 0.2 ^e^	* 97 ± 1.78 ^e^	94 ± 0.76 ^f^		
Punjab	* 59 ± 2.8 ^e^	24 ± 0.4 ^f^	* 11.7 ± 1.2 ^gh^	5.6 ± 0.4 ^de^	* 104 ± 1.52 ^d^	94 ± 0.76 ^f^		
LSD 5%	2.22		0.82		1.0			

Mean values with same alphabets (a, b, c, d………) in superscript in a column do not differ significantly. Mean with * showing the significant effect of water deficit stress against each genotype. PLA = plant leaf area; FLA = flag leaf area; RFW = root fresh weight; SFW = shoot fresh weight; RDW = root dry weight; SDW = shoot dry weight; PL = plant length.

**Table 2 plants-11-00466-t002:** Yield parameters of newly developed wheat genotypes grown under water deficit stress and normal irrigation (Mean ± SE; *n* = 4).

	NOT/Plant		FT/Plant		SPK L (cm)		Spkt/Spk		STR Y (g)	
Genotypes	Normal Irrigation	Water Deficit Stress	Normal Irrigation	Water Deficit Stress	Normal Irrigation	Water Deficit Stress	Normal Irrigation	Water Deficit Stress	Normal Irrigation	Water Deficit Stress
MP1	* 40 ± 1.8 ^b^	18 ± 1.5 ^c^	* 34 ± 1.8 ^a^	13 ± 1.5 ^c^	* 14.5 ± 0.3 ^f^	13.3 ± 0.3 ^e^	* 21.3 ± 0.3 ^d^	20.0 ± 0.6 ^cde^	* 52.2 ± 1.8 ^d^	30.8 ± 1.5 ^a^
MP2	* 28 ± 2.5 ^f^	18 ± 0.6 ^c^	* 24 ± 1.1 ^e^	12 ± 2.7 ^cd^	* 15.0 ± 0.0 ^e^	14.5 ± 0.3 ^bc^	* 22.0 ± 0.1 ^c^	20.3 ± 0.3 ^c^	* 51.4 ± 2.8 ^de^	24.2 ± 0.7 ^c^
MP 3	* 34 ± 2.0 ^cd^	23 ± 1.4 ^a^	* 29 ± 2.0 ^c^	21 ± 3.5 ^a^	* 17.2 ± 0.4 ^a^	15.3 ± 0.7 ^c^	* 21.3 ± 0.9 ^d^	20.3 ± 0.3 ^c^	* 50.0 ± 2.5 ^e^	28.5 ± 2.2 ^b^
MP 4	* 34 ± 2.0 ^cd^	14 ± 1.0 ^e^	* 31 ± 1.5 ^b^	12 ± 1.0 ^cd^	* 16.7 ± 0.3 ^b^	15.7 ± 0.3 ^a^	* 23.3 ± 0.7 ^b^	21.3 ± 0.3 ^a^	* 58.1 ± 3.2 ^a^	20.2 ± 1.2 ^e^
MP 5	* 44 ± 1.8 ^a^	12 ± 0.9 ^f^	* 35 ± 0.9 ^a^	10 ± 1.5 ^d^	* 16.5 ± 0.3 ^bc^	14.3 ± 0.3 ^c^	21.0 ± 0.6 ^edf^	20.7 ± 0.7 ^b^	* 56.3 ± 0.8 ^b^	24.0 ± 1.2 ^c^
MP 6	* 33 ± 2.2 ^d^	15 ± 0.7 ^de^	* 30 ± 2.0 ^bc^	13 ± 1.2 ^c^	14.5 ± 0.5 ^f^	14.5 ± 0.3 ^bc^	* 24.0 ± 0.6 ^a^	21.3 ± 0.7 ^a^	* 48.4 ± 1.6 ^f^	20.3 ± 1.4 ^e^
MP 7	* 35 ± 2.3 ^c^	15. ± 1.2 ^de^	*28 ± 1.5 ^cd^	13 ± 0.6 ^c^	* 16.2 ± 0.4 ^c^	14.5 ± 0.5 ^bc^	19.7 ± 0.3 ^g^	19.7 ± 0.3 ^e^	* 38.5 ± 1.1 ^h^	18.7 ± 1.7 ^e^
MP 8	* 26 ± 0.3 ^g^	14 ± 0.9 ^e^	* 24 ± 0.3 ^e^	11 ± 1.2 ^d^	* 16.5 ± 0.5 ^bc^	14.7 ± 0.3 ^b^	20.3 ± 0.3 ^f^	19.7 ± 0.3 ^b^	* 32.5 ± 1.3 ^j^	22.3 ± 1.5 ^d^
MP 9	* 30 ± 1.4 ^e^	16 ± 0.8 ^d^	* 27 ± 2.3 ^d^	15 ± 0.3 ^b^	* 16.0 ± 0.5 ^c^	14.7 ± 0.3 ^b^	* 21.7 ± 0.3 ^c^	20.7 ± 0.3 ^b^	* 36.1 ± 2.0 ^i^	23.7 ± 2.0 ^cd^
MP 10	* 30 ± 1.7 ^e^	21 ± 1.5 ^b^	* 26 ± 1.3 ^d^	12 ± 2.6 ^cd^	14.0 ± 0.0 ^g^	13.9 ± 0.2 ^d^	20.7 ± 0.3 ^e^	20.7 ± 0.3 ^b^	* 42.9 ± 2.2 ^g^	29.0 ± 2.0 ^b^
Millat	* 34 ± 0.9 ^cd^	14 ± 0.9 ^e^	* 27 ± 0.9 ^d^	13 ± 0.6 ^c^	* 14.5 ± 0.1 ^f^	13.0 ± 0.0 ^e^	* 21.7 ± 0.3 ^c^	20.0 ± 0.6 ^cde^	* 50.7 ± 2.2 ^de^	22.3 ± 0.9 ^d^
Punjab	* 35 ± 0.6 ^c^	14 ± 0.9 ^e^	* 27 ± 0.9 ^d^	12 ± 1.2 ^cd^	* 15.5 ± 0.5 ^d^	13.9 ± 0.4 ^d^	* 22.0 ± 0.6 ^c^	21.0 ± 0.6 ^a^	* 54.0 ± 1.5 ^c^	22.3 ± 2.3 ^d^
LSD 5%	1.40		1.50		0.31		0.39		1.62	
	NOG/Spk		GW/Spk (g)		100 GW (g)		GY/Plant (g)			
Genotypes	Normal irrigation	Water deficit stress	Normal irrigation	Water deficit stress	Normal irrigation	Water deficit stress	Normal irrigation	Water deficit stress		
MP1	* 62.7 ± 4.1 ^de^	42.0 ± 2.0 ^d^	* 2.6 ± 0.2 ^b^	1.3 ± 0.1 ^f^	* 4.2 ± 0.1^a^	3.2 ± 0.1 ^d^	* 38.9 ± 1.1 ^a^	26.7 ± 0.9^a^		
MP2	* 56.7 ± 2.4 ^f^	39.0 ± 3.0 ^e^	* 2.2 ± 0.1 ^e^	1.2 ± 0.1 ^g^	* 3.9 ± 0.1 ^c^	3.0 ± 0.1 ^f^	* 26.7 ± 0.3 ^f^	16.3 ± 1.2 ^g^		
MP 3	* 66.3 ± 2.3 ^c^	45.7 ± 1.2 ^b^	* 2.4 ± 0.1 ^d^	1.3 ± 0.1 ^f^	* 3.7 ± 0.1 ^d^	2.9 ± 0.1 ^g^	* 34.0 ± 0.6 ^c^	22.0 ± 2.1 ^c^		
MP 4	* 67.3 ± 2.2 ^c^	37.3 ± 1.2 ^f^	* 2.4 ± 0.1 ^d^	1.0 ± 0.1 ^i^	* 3.5 ± 0.1 ^e^	2.7 ± 0.1 ^h^	* 26.3 ± 1.9 ^f^	18.0 ± 1.0 ^f^		
MP 5	* 72.7 ± 2.2 ^b^	44.7 ± 1.7 ^c^	* 2.7 ± 0.1 ^a^	1.4 ± 0.1 ^e^	* 3.7 ± 0.1 ^d^	3.1 ± 0.1 ^e^	* 27.2 ± 1.1 ^e^	23.5 ± 1.1 ^b^		
MP 6	* 67.3 ± 2.0 ^c^	35.0 ± 2.5 ^g^	* 2.4 ± 0.1 ^d^	0.8 ± 0.1 ^j^	* 3.5 ± 0.1 ^e^	2.2 ± 0.2 ^j^	* 23.0 ± 1.2 ^g^	13.3 ± 0.3 ^i^		
MP 7	* 60.0 ± 0.6 ^e^	40.7 ± 2.0 ^de^	* 2.1 ± 0.1 ^f^	1.1 ± 0.1 ^h^	* 3.5 ± 0.1 ^e^	2.7 ± 0.1 ^h^	* 28.1 ± 1.4 ^d^	20.6 ± 1.4 ^e^		
MP 8	* 73.7 ± 1.3 ^ab^	53.7 ± 2.4 ^a^	* 2.7 ± 0.1 ^a^	1.8 ± 0.1 ^a^	* 3.7 ± 0.1 ^d^	3.3 ± 0.2 ^c^	* 27.7 ± 0.7 ^e^	21.2 ± 2.1 ^d^		
MP 9	* 75.0 ± 2.5 ^a^	41.7 ± 2.7 ^d^	* 2.5 ± 0.1 ^c^	1.1 ± 0.1 ^h^	* 3.3 ± 0.1 ^f^	2.5 ± 0.2 ^i^	* 23.0 ± 1.0 ^g^	14.3 ± 1.5 ^h^		
MP 10	* 63.0 ± 1.5 ^d^	46.3 ± 2.7 ^b^	* 2.5 ± 0.1 ^c^	1.7 ± 0.1 ^b^	* 4.0 ± 0.2 ^b^	3.6 ± 0.1 ^a^	* 27.0 ± 1.1 ^e^	20.3 ± 1.2 ^e^		
Millat	* 61.0 ± 2.9 ^e^	47.7 ± 1.5 ^b^	* 2.4 ± 0.1 ^d^	1.6 ± 0.1 ^c^	* 4.0 ± 0.1 ^b^	3.3 ± 0.1 ^c^	* 26.3 ± 0.7 ^f^	16.3 ± 0.7 ^g^		
Punjab	* 58.3 ± 1.1 ^f^	44.3 ± 2.7 ^c^	* 2.3 ± 0.2 ^e^	1.5 ± 0.1 ^d^	* 3.9 ± 0.1 ^c^	3.5 ± 0.1 ^b^	* 35.3 ± 0.9 ^b^	23.0 ± 0.6 ^b^		
LSD 5%	1.94		0.76		0.09		0.97			

Mean values with same alphabets (a, b, c, d………) in superscript in a column do not differ significantly. Mean with * showing the significant effect of water deficit stress against each genotype. FT/plant = fertile tillers per plant; NOT/plant = number of tillers per plant; Spk L = spike length; Spkt/Spk = number of spiklets per spike; STR Y = straw yield per plant; 100 GW = 100 grain weight; GW/Spk = grain weight per spike; NOG/Spk = number of grains per spike; GY/Plant = grain yield per plant.

**Table 3 plants-11-00466-t003:** Leaf photosynthetic pigments and LRWC of newly developed wheat genotypes when grown water deficit stress and normal irrigation (Mean ± SE; *n* = 4).

	Chl. *a* (mg/g DW)		Chl. *b* (mg/g DW)		Chl. *a/b*		T. Chl. (mg/g DW)	
Genotypes	Normal Irrigation	Water Deficit Stress	Normal Irrigation	Water Deficit Stress	Normal Irrigation	Water Deficit Stress	Normal Irrigation	Water Deficit Stress
MP 1	* 4.40 ± 0.05 ^i^	4.15 ± 0.07 ^i^	* 2.53 ± 0.06 ^d^	2.80 ± 0.06 ^d^	1.74 ± 0.2 ^d^	1.48 ± 0.2 ^g^	6.92 ± 0.01 ^i^	6.95 ± 0.01 ^g^
MP 2	* 4.39 ± 0.09 ^i^	4.72 ± 0.03 ^g^	* 2.39 ± 0.06 ^e^	2.53 ± 0.06 ^f^	1.84 ± 0.3 ^d^	1.87 ± 0.1 ^def^	* 6.78 ± 0.01 ^j^	7.25 ± 0.04 ^e^
MP 3	* 6.21 ± 0.02 ^b^	5.38 ± 0.03 ^c^	3.29 ± 0.10 ^a^	3.26 ± 0.06 ^c^	1.89 ± 0.3 ^d^	1.65 ± 0.1 ^fg^	* 9.50 ± 0.05 ^a^	8.63 ± 0.01 ^c^
MP 4	* 6.30 ± 0.10 ^a^	4.53 ± 0.06 ^h^	* 2.94 ± 0.03 ^b^	1.46 ± 0.03 ^k^	* 2.14 ± 0.2 ^c^	3.11 ± 0.2 ^a^	* 9.23 ± 0.02 ^b^	5.99 ± 0.03 ^h^
MP 5	* 4.98 ± 0.09 ^d^	5.11 ± 0.06 ^d^	* 2.10 ± 0.03 ^h^	2.01 ± 0.09 ^h^	2.37 ± 0.3 ^c^	2.54 ± 0.3 ^bc^	7.08 ± 0.02 ^h^	7.12 ± 0.05 ^f^
MP 6	* 5.41 ± 0.02 ^e^	4.28 ± 0.07 ^i^	* 2.35 ± 0.03 ^e^	1.59 ± 0.09 ^j^	* 2.31 ± 0.1 ^c^	2.71 ± 0.2 ^b^	* 7.75 ± 0.01 ^f^	5.86 ± 0.06 ^h^
MP 7	* 5.27 ± 0.08 ^f^	4.04 ± 0.10 ^k^	* 2.90 ± 0.09 ^bc^	1.72 ± 0.03 ^i^	* 1.82 ± 0.2 ^d^	2.35 ± 0.2 ^c^	* 8.16 ± 0.06 ^d^	5.75 ± 0.03 ^f^
MP 8	* 5.75 ± 0.04 ^d^	7.08 ± 0.01 ^b^	* 2.94 ± 0.03 ^b^	4.24 ± 0.06 ^a^	1.95 ± 0.1 ^d^	1.67 ± 0.2 ^fg^	* 8.69 ± 0.01 ^c^	11.33 ± 0.08 ^b^
MP 9	* 5.40 ± 0.05 ^e^	4.99 ± 0.04 ^e^	2.23 ± 0.03 ^fg^	2.27 ± 0.03 ^g^	2.42 ± 0.2 ^bc^	2.20 ± 0.2 ^cd^	* 7.63 ± 0.02 ^g^	7.27 ± 0.03 ^e^
MP 10	* 5.20 ± 0.03 ^g^	4.92 ± 0.01 ^f^	* 2.86 ± 0.09 ^c^	2.67 ± 0.12 ^e^	1.82 ± 0.3 ^d^	1.84 ± 0.3 ^ef^	* 8.07 ± 0.02 ^de^	7.59 ± 0.03^d^
Millat	* 5.85 ± 0.08 ^c^	8.26 ± 0.08 ^a^	* 2.17 ± 0.09 ^g^	3.39 ± 0.03 ^b^	2.70 ± 0.5 ^b^	2.43 ± 0.2 ^bc^	* 8.01 ± 0.03 ^e^	11.66 ± 0.03 ^a^
Punjab	* 5.40 ± 0.10 ^e^	4.93 ± 0.02 ^ef^	* 1.48 ± 0.06 ^i^	2.60 ± 0.03 ^e^	* 3.65 ± 0.2 ^a^	1.90 ± 0.5 ^def^	* 6.88 ± 0.06 ^i^	7.53 ± 0.07 ^d^
LSD 5%	0.06		0.07		0.35		0.10	
	Car. (µg/g DW)		T. Chl./Car		LRWC (%)			
Genotypes	Normal irrigation	Water deficit stress	Normal irrigation	Water deficit stress	Normal irrigation	Water deficit stress		
MP 1	* 53.3 ± 1.9 ^bc^	58.6 ± 1.0 ^ab^	* 45.2 ± 0.1 ^b^	43.1 ± 0.1 ^abc^	* 80.3 ± 2.6 ^def^	73.0 ± 1.5 ^bc^		
MP 2	54.9 ± 1.8 ^b^	56.0 ± 0.5 ^de^	43.4 ± 0.1 ^de^	42.4 ± 0.1 ^bc^	* 82.3 ± 0.8 ^ab^	70.0 ± 2.1 ^f^		
MP 3	* 57.0 ± 1.6 ^a^	59.6 ± 1.6 ^a^	41.5 ± 0.1 ^f^	42.1 ± 0.1 ^c^	* 82.0 ± 0.7 ^abc^	70.2 ± 1.0 ^ef^		
MP 4	52.8 ± 1.1 ^c^	53.7 ± 1.1 ^f^	* 43.6 ± 0.1 ^cd^	39.4 ± 0.1 ^d^	* 83.0 ± 1.6 ^a^	70.7 ± 0.7 ^ef^		
MP 5	* 52.4 ± 2.2 ^c^	59.9 ± 1.5 ^a^	* 42.6 ± 0.1 ^ef^	37.7 ± 0.0 ^ef^	* 80.1 ± 2.2 ^efg^	74.3 ± 0.7 ^b^		
MP 6	* 47.3 ± 1.8 ^d^	58.1 ± 0.5 ^abc^	* 47.1 ± 0.1 ^a^	37.4 ± 0.2 ^f^	* 82.2 ± 0.8 ^ab^	70.0 ± 1.0 ^f^		
MP 7	54.1 ± 0.6 ^bc^	55.1 ± 0.9 ^ef^	* 43.9 ± 0.1 ^cd^	39.5 ± 0.1 ^d^	* 79.8 ± 0.9 ^efg^	69.6 ± 1.6 ^f^		
MP 8	* 52.12 ± 0.5 ^c^	57.6 ± 1.5 ^bcd^	* 45.2 ± 0.1 ^b^	43.7 ± 0.2 ^a^	* 81.4 ± 0.9 ^bcd^	74.7 ± 0.9 ^a^		
MP 9	* 52.8 ± 2.2 ^c^	58.0 ± 0.8 ^abc^	* 41.6 ± 0.1 ^f^	39.7 ± 0.2 ^d^	* 78.9 ± 0.7 ^g^	69.7 ± 1.f ^e^		
MP 10	* 53.6 ± 0.6 ^bc^	58.1 ± 0.7 ^abc^	* 44.6 ± 0.2 ^bc^	43.5 ± 0.1 ^ab^	* 80.9 ± 1.5 ^cde^	74.7 ± 0.9 ^a^		
Millat	55.3 ± 1.7 ^ab^	55.2 ± 1.0 ^ef^	* 40.1 ± 0.0 ^g^	38.6 ± 0.1 ^de^	* 79.4 ± 2.1 ^fgf^	71.3 ± 2.7 ^de^		
Punjab	* 53.2 ± 1.3 ^bc^	56.3 ± 2.0 ^cde^	* 39.1 ± 0.1 ^g^	28.4 ± 0.0 ^g^	* 81.5 ± 1.6 ^bc^	72.0 ± 0.6 ^cd^		
LSD 5%	1.9		1.10		1.20			

Mean values with same alphabets (a, b, c, d………) in superscript in a column do not differ significantly. Mean with * showing the significant effect of water deficit stress against each genotype. Chl. *a* = leaf chlorophyll *a*; Chl. *B* = leaf chlorophyll *b*; Chl. *a*/*b* = leaf chlorophyll a/chlorophyll *b*; T. Chl. = Leaf total chlorophyll; Car. = leaf carotenoid; T. Chl./Car = leaf total chlorophyll/leaf carotenoid; LRWC = leaf relative water content.

**Table 4 plants-11-00466-t004:** Different stress tolerance indices of newly developed wheat genotypes under limited water supply and normal irrigation (Mean ± SE; *n* = 4).

Genotypes	SSI	STI	GMP	MP	TOL	HAM	YI	YSI	DI
MP1	1.003	1.266	32.220	32.800	12.26	31.653	0.938	0.685	0.930
MP2	1.233	0.531	20.860	21.500	10.33	20.258	0.574	0.613	0.509
MP3	1.123	0.912	27.340	28.000	12.00	26.714	0.774	0.647	0.725
MP4	1.007	0.578	21.770	22.167	8.333	21.383	0.633	0.684	0.627
MP5	0.433	0.778	25.240	25.317	3.700	25.181	0.825	0.864	1.032
MP6	1.338	0.374	17.510	18.167	9.667	16.881	0.469	0.580	0.394
MP7	0.848	0.708	24.090	24.383	7.500	23.807	0.726	0.733	0.771
MP8	0.744	0.715	24.210	24.433	6.467	24.005	0.746	0.766	0.827
MP9	1.199	0.402	18.150	18.667	8.667	17.661	0.504	0.623	0.455
MP10	0.789	0.670	23.440	23.683	6.700	23.209	0.715	0.752	0.779
Millat	1.209	0.525	20.630	21.333	10.000	20.161	0.574	0.620	0.516
Punjab	1.111	0.991	28.500	29.167	12.333	27.863	0.809	0.651	0.762

**Table 5 plants-11-00466-t005:** Spearman’s correlation coefficient values (*r^2^*) of yield and growth parameters with biochemical and physiological parameters of newly developed wheat inbreed lines grown in limited water supply with normal irrigation.

	FLA	NOL	PLA	SFW	RFW	SDW	RDW	PL	NOT	FT	SPK L	NO Spkt/Spk	NOG/Spk	GW/Spk	100 GW	GY/Plant
FLA	1.000 ***															
NOL	0.742 ***	1.000 ***														
PLA	0.924 ***	0.920 ***	1.000 ***													
SFW	0.865 ***	0.878 ***	0.917 ***	1.000 ***												
RFW	0.842 ***	0.866 ***	0.923 ***	0.932 ***	1.000 ***											
SDW	0.821 ***	0.814 ***	0.854 ***	0.940 ***	0.894 ***	1.000 ***										
RDW	0.769 ***	0.789 ***	0.838 ***	0.888 ***	0.874 ***	0.874 ***	1.000 ***									
PL	0.535 ***	0.473 ***	0.533 ***	0.533 ***	0.483 ***	0.567 ***	0.496 ***	1.000 ***								
NOT	0.749 ***	0.596 ***	0.691 ***	0.831 ***	0.745 ***	0.859 ***	0.765 ***	0.530 ***	1.000 ***							
FT	0.777 ***	0.642 ***	0.726 ***	0.858 ***	0.777 ***	0.879 ***	0.795 ***	0.535 ***	0.966 ***	1.000 ***						
SPK L	0.613 ***	0.45 8***	0.596 ***	0.553 ***	0.591 ***	0.543 ***	0.616 ***	0.355 **	0.473 ***	0.527 ***	1.000 ***					
NO Spkt/Spk	0.295 **	0.324 **	0.312 **	0.402 ***	0.474 ***	0.377 **	0.476 ***	0.158 ns	0.337 **	0.359 **	0.253 *	1.000 ***				
NOG/Spk	0.830 ***	0.608 ***	0.746 ***	0.790 ***	0.726 ***	0.765 ***	0.709 ***	0.552 ***	0.754 ***	0.770 ***	0.536 ***	0.331 **	1.000 ***			
GW/Spk	0.833 ***	0.563 ***	0.719 ***	0.779 ***	0.698 ***	0.755 ***	0.679 ***	0.543 ***	0.803 ***	0.789 ***	0.433 ***	0.333 **	0.949 ***	1.000 ***		
100 GW	0.641 ***	0.348 **	0.502 ***	0.580 ***	0.488 ***	0.545 ***	0.455 ***	0.368 ns	0.676 ***	0.610 ***	0.133 ns	0.246 *	0.658 ***	0.855 ***	1.000 ***	
GY/plant	0.691 ***	0.552 ***	0.643 ***	0.686 ***	0.583 ***	0.662 ***	0.604 ***	0.476 ***	0.701 ***	0.665 ***	0.316 **	0.163 ns	0.589 ***	0.716 ***	0.759 ***	1.000 ***
LRMP	−0.808 ***	−0.644 ***	−0.745 ***	−0.841 ***	−0.771 ***	−0.831 ***	−0.750 ***	−0.619 ***	−0.834 ***	−0.812 ***	−0.467 ***	−0.325 **	−0.791 ***	−0.839 ***	−0.723 ***	−0.729 ***
LRWC	0.791 ***	0.594 ***	0.721 ***	0.790 ***	0.734 ***	0.750 ***	0.706 ***	0.519 ***	0.736 ***	0.719 ***	0.406 ***	0.486 ***	0.783 ***	0.825 ***	0.711 ***	0.686 ***
Chl. *a.*	−0.055 ns	−0.197 ns	−0.105 ns	−0.110 ns	0.014 ns	−0.047 ns	0.021 ns	−0.015 ns	0.047 ns	−0.007 ns	0.195 ns	0.083 ns	−0.018 ns	0.003 ns	0.000 ns	−0.045 ns
Chl. *b.*	0.049 ns	0.118 ns	0.109 ns	0.091 ns	0.004 ns	0.062 ns	0.058 ns	0.389 ***	0.036 ns	0.029 ns	−0.050 ns	−0.274 *	0.056 ns	0.069 ns	0.114 ns	0.154 ns
T. Chl.	0.074 ns	0.059 ns	0.100 ns	0.093 ns	0.073 ns	0.125 ns	0.144 ns	0.445 ***	0.132 ns	0.111 ns	0.090 ns	−0.189 ns	0.103 ns	0.098 ns	0.070 ns	0.117 ns
Chl. *a/b*	−0.078 ns	−0.175 ns	−0.155 ns	−0.132 ns	−0.035 ns	−0.092 ns	−0.098 ns	−0.371 **	−0.036 ns	−0.052 ns	0.050 ns	0.252 *	−0.114 ns	−0.100 ns	−0.094 ns	−0.076 ns
Car	−0.489 ***	−0.425 ***	−0.452 ***	−0.485 ***	−0.458 ***	−0.520 ***	−0.367 **	−0.252 *	−0.458 ***	−0.487 ***	−0.283 *	−0.418 ***	−0.533 ***	−0.536 ***	−0.390 ***	−0.227 ***
T. Chl./Car	0.385 ***	0.338 **	0.389 ***	0.401 ***	0.370 **	0.448 ***	0.370 **	0.553 ***	0.414 ***	0.415 ***	0.260 *	0.121 ns	0.435 ***	0.431 ***	0.309 **	0.245 *
TPC	−0.329 **	−0.241 **	−0.299 *	−0.332 **	−0.283 *	−0.266 ns	−0.200 ns	−0.015 ns	−0.339 **	−0.303 **	−0.154 ns	−0.105 ns	−0.355 **	−0.367 ***	−0.341 **	−0.111 ns
FAA	−0.630 ***	−0.493 ***	−0.596 ***	−0.649 ***	−0.704 ***	−0.700 ***	−0.748 ***	−0.529 ***	−0.624 ***	−0.662 ***	−0.569 ***	−0.553 ***	−0.613 ***	−0.589 ***	−0.398 ***	−0.416 **
TSP	−0.126 ns	−0.247 *	−0.262 *	−0.177 ns	−0.248 *	−0.238 *	−0.282 *	−0.506 ***	−0.056 ns	−0.103 ns	−0.412 ***	0.000 ns	−0.147 ns	−0.050 ns	0.134 ns	0.016 ns
Antho	0.005 ns	−0.001 ns	0.054 ns	−0.096 ns	−0.074 ns	0.004 ns	0.005 ns	0.286 *	−0.170 ns	−0.135 ns	0.164	−0.262 *	0.009 ns	−0.063 ns	−0.173 ns	−0.074 ns
POD	0.058 ns	0.145 ns	0.173 ns	0.063 ns	0.121 ns	0.051 ns	0.083 ns	0.172 ns	−0.048 ns	−0.014 ns	0.082 ns	−0.148 ns	0.023 ns	0.004 ns	−0.035 ns	0.112 ns
SOD	−0.311 **	−0.307 **	−0.289 *	−0.366 **	−0.315 **	−0.466 ***	−0.492 ***	−0.223 ns	−0.444 ***	−0.436 ***	−0.380 **	−0.152 ns	−0.239 *	−0.244 *	−0.148 ns	−0.297 *
APX	−0.214 ns	−0.244 *	−0.208 ns	−0.315 **	−0.288 *	−0.325 **	−0.210 ns	0.071 ns	−0.290 *	−0.278 *	−0.228 *	−0.189 ns	−0.103 ns	−0.117 ns	−0.125 ns	−0.101 ns
CAT	−0.120 ns	−0.062 ns	−0.098 ns	−0.114 ns	−0.108 ns	−0.108 ns	−0.075 ns	0.003 ns	−0.078 ns	−0.085 ns	0.000 ns	−0.011 ns	−0.071 ns	−0.127 ns	−0.180 ns	−0.135 ns
H_2_O_2_	−0.688 ***	−0.584 ***	−0.663 ***	−0.720 ***	−0.649 ***	−0.652 ***	−0.619 ***	−0.570 ***	−0.64 0***	−0.643 ***	−0.274 *	−0.383 ***	−0.797 ***	−0.818 ***	−0.681 ***	−0.507 ***
MDA	−0.666 ***	−0.458 ***	−0.603 ***	−0.643 ***	−0.627 ***	−0.709 ***	−0.633 ***	−0.568 ***	−0.757 ***	−0.742 ***	−0.472 ***	−0.292 ns	−0.652 ***	−0.673 ***	−0.502 ***	−0.533 ***
AsA	−0.775 ***	−0.637 ***	−0.728 ***	−0.786 ***	−0.787 ***	−0.824 ***	−0.773 ***	−0.559 ***	−0.754 ***	−0.780 ***	−0.606 ***	−0.440 ***	−0.721 ***	−0.725 ***	−0.541 ***	−0.659 ***
Flav	−0.361 ***	−0.540 ***	−0.432 ***	−0.446 ***	−0.478 ***	−0.480 ***	−0.441 ***	−0.170 ns	−0.257 *	−0.364 **	−0.365 **	−0.311 **	−0.290 ns	−0.228 *	−0.016 ns	−0.101 ns
TSS	−0.721 ***	−0.595 ***	−0.668 ***	−0.738 ***	−0.731 ***	−0.760 ***	−0.659 ***	−0.550 ***	−0.678 ***	−0.721 ***	−0.510 ***	−0.325 **	−0.758 ***	−0.720 ***	−0.499 ***	−0.467 ***
RS	−0.659 ***	−0.541 ***	−0.605 ***	−0.667 ***	−0.632 ***	−0.690 ***	−0.582 ***	−0.559 ***	−0.643 ***	−0.663 ***	−0.490 ***	−0.273 *	−0.649 ***	−0.628 ***	−0.453 ***	−0.494 ***
NRS	−0.713 ***	−0.590 ***	−0.662 ***	−0.731 ***	−0.728 ***	−0.753 ***	−0.654 ***	−0.538 ***	−0.668 ***	−0.713 ***	−0.502 ***	−0.325 **	−0.755 ***	−0.716 ***	−0.494 ***	−0.455 ***

*, ** and *** = significant at 0.05, 0.01 and 0.001 levels, respectively. ns = non-significant.3.11. Principal Component Analysis (PCA) Analysis of Varying Studied Parameters of Wheat Genotypes Grown under Limited Water Supply and Normal Irrigation.

## Data Availability

The data presented in the manuscript is the sole data and no other data is linked with this data. The data presented in this study is available in supplementary material.

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
