# Peer review of "Water Deficit Stress Tolerance Potential of Newly Developed Wheat Genotypes for Better Yield Based on Agronomic Traits and Stress Tolerance Indices: Physio-Biochemical Responses, Lipid Peroxidation and Antioxidative Defense Mechanism"

_plants, 2022, doi:10.3390/plants11030466_

Round 1

Reviewer 1 Report

The authors detected the effects of water stress on newly developed wheat genotypes. I think results are very valuable. However, this manuscript passes better to an agricultural journal such as Agronomy.  Aims and Material and Methods are clear and up-to-date. I think the only discussion is weak. Moreover, the presentation of results has to be improved.

1. Please add an accurate description with the abbreviation to the tables and figures. 2. Results are the men of the result from the 2 different years? Please answer it in 2.17. Statistical analysis subchapter. 3. Please order newly and add a letter to Fig 1: A H2O2, B: SOD, C: CAT, D: APX, E: POD, F: MDA, G: LRMP. Add correct axis x with the name of the genotypes. Please mark with * the most significant differences between non-stressed and stressed conditions. 4. In the discussion please write about the basic antioxidant capacity and the changes in this in the different genotypes more carefully. 5. What is the relation between the enzymatic and non-enzymatic antioxidants between the genotypes. What is the capacity and potential to change under stressed conditions?  6. Photosynthetic activity was not measured but there are many mentions in the discussion. Please discuss more carefully the basic levels and changes in sugar content and photosynthetic pigment levels.

Author Response

Response to Reviewer 1 comments

Comment: 1. The authors detected the effects of water stress on newly developed wheat genotypes. I think results are very valuable.

Response: Thanks the worth reviewer for nice comments

Comment: Aims and Material and Methods are clear and up-to-date.

Response: Thanks the worthy for the appreciation

Comment:. I think the only discussion is weak.

Response: Discussion section is improved and rewritten

Comment: Moreover, the presentation of results has to be improved.

Response: Presentation of the results has been improved

Comment:Please add an accurate description with the abbreviation to the tables and figures.

Response: Now the description regarding the abbreviations has been presented clearly in tables and figures. The addition regarding the abbreviations is presented in red text with each table and figure  

Comment: Results are the men of the result from the 2 different years? Please answer it in 2.17. Statistical analysis sub chapter.

Response: The required information is incorporated in sub chapter 2.18 “statistical analysis ” Lines 339-341

Comment: Please order newly and add a letter to Fig 1: A H2O2, B: SOD, C: CAT, D: APX, E: POD, F: MDA, G: LRMP. Add correct axis x with the name of the genotypes. Please mark with * the most significant differences between non-stressed and stressed conditions.

Response: Suggestion is followed regarding the amendments in Fig 1. X-axis has been corrected with the names of wheat genotypes in all Figs. The most significant differences between non-stressed and water deficit stressed in figures on the bars and in tables with mean values has been marked with *. During editing any elaborated differences has been incorporated in results part too

Comment: In the discussion please write about the basic antioxidant capacity and the changes in this in the different genotypes more carefully. What is the relation between the enzymatic and non-enzymatic antioxidants between the genotypes. What is the capacity and potential to change under stressed conditions? 

Response: Now the required information regarding basic antioxidant capacity and changes that took place in genotypes has been incorporated in discussion part of the manuscript. Please see lines 713-783

Comment: Photosynthetic activity was not measured but there are many mentions in the discussion. Please discuss more carefully the basic levels and changes in sugar content and photosynthetic pigment levels.

Response: Suggestion is followed and the discussion part is improved in view of the comments of the worthy reviewer. Basic information has been incorporated in the discussion part in relation with sugar contents, and photosynthetic pigments as well as the in relation with the comparative responses of genotypes under water deficit stress

Reviewer 2 Report

The aim of this work was to select drought tolerant wheat inbreeds lines using several morpho-physio-biochemical, agronomic and yield characteristics. Drought is undoubtedly one of the most challenging environmental stress factor that affect plant growth, development, and productivity worldwide. Therefore, it is very important understand the impact of this stress on plant performance and select drought tolerant species.

The introduction is well structured, and the authors selected appropriated methods to conduct this experiment. However, several important points in this work must be improved.

The English must be revised by a native or a professional. The quality of the write (English) is quite low, making difficult to understand what the authors wants to describe.

The page and line numbers are not provided.

Introduction:

For example, the 2nd paragraph: “The plants species with better performance in all or few on above mentioned mechanism are categorized as drought tolerant are sensitive ones”.

The introduction must be more focussed on the physiological and biochemical responses to drought and not so vague.

M&M:

-Water stressed: please replace this word by water deficit stress. “Water stressed” can mean both water deficit or water in excess.

-The intensity of water deficit is not mentioned. How many water/irrigation was add in the plants exposed to water deficit conditions?

- Meteorological data, like temperature and humidity during the experiment, is also not provided. In a field experiment several environmental factors are  present and can affect the assay, so it is necessary to take in to consideration some of these weather condition to  better discuss and make conclusion. 

-Sampling time is not referred? When where the samples collected?

-Number of plants used, and number of plants /treatment?

-Be careful with the units: 15vm on pg.3

- In the section “2.15". Estimation of Stress tolerance indices” The description of the meaning af all formulae used must be provided in this section. For example in the section 2.15.1, the formulae/indices: SSI=1-(Ys/Yp)/SI. What is the meaning of the Ys, Yp and SI? This must be provide in this section and not at the end of the manuscript. 

Results:

- For more accuracy, all data should be expressed per leaf dry weight, and not per leaf fresh weight, since there is a change in the leaf relative water content. This will influence the levels of some parameters, such as pigments. Also, the description of the results must follow the statistical data. We can not say that something decreased if it is not significant. This must be carefully revised.

- The quality of Figures 1 and 2 must be improved. Bars seems to be overlapped.

Discussion:

In general the discussion is too long, several parts are repeated, is very confuse and seems more a comparison and presentation of data from other studies. In sum, there is a need for more discussion (not in word count but in quality). How the observed morpho-,physio-, biochemical-, and agronomic changes are related to each other and with the stress imposed?  Also the discussion must be supported by the obtained data and not with the available literature. Some parts of the discussion are expendable, eg the first 2 paragraphs.  The authors failed to give a proper concluding message for the readers.

Author Response

Response to Reviewer 2 comments

The introduction is well structured, and the authors selected appropriated methods to conduct this experiment. However, several important points in this work must be improved.

Response: Thanks the worthy reviewer for the nice comments. The technical comments are also very useful in improving the manuscript

Comment: 1. The English must be revised by a native or a professional. The quality of the write (English) is quite low, making difficult to understand what the authors wants to describe.

Response: Suggestion is followed and the English grammer and composition has been improved by a native expert. The addition and corrections are incorporation in red text through out the manuscipt

Comment: 3. The page and line numbers are not provided.

Response: Now the page and line numbers has been incorporated

Introduction:

Comment:. For example, the 2nd paragraph: “The plants species with better performance in all or few on above mentioned mechanism are categorized as drought tolerant are sensitive ones”. The introduction must be more focused on the physiological and biochemical responses to drought and not so vague.

Response: Suggestion is followed and the Introduction part has been improved and the added information is in red text please see lines 64-73

M&M:

Comment: 1-Water stressed: please replace this word by water deficit stress. “Water stressed” can mean both water deficit or water in excess.

Response: Thanks for the nice comments. The suggestion is followed and the water stressed is replaced with water deficit stress in whole the manuscript, including the Figures, and Tables.  

Comment: 2-The intensity of water deficit is not mentioned. How many water/irrigation was add in the plants exposed to water deficit conditions?

Response: the information is incorporated in the text at lines 129-131

Comment: 3. - Meteorological data, like temperature and humidity during the experiment, is also not provided. In a field experiment several environmental factors are  present and can affect the assay, so it is necessary to take in to consideration some of these weather condition to  better discuss and make conclusion.

Response: Meteorological data during the experimental period has been added in the text in section 2.2 of the materials and methods. Please see lines 153-158  

Comment: 4. -Sampling time is not referred? When where the samples collected? Number of plants used, and number of plants /treatment?

Response: Sampling time and method of sampling is incorporated in text at lines 148-151

Comment: 5-Be careful with the units: 15vm on pg.3

Response: correction is incorporated and other corrections are also made in the text in red regarding the units presentation

Comment: 7- In the section “2.15". Estimation of Stress tolerance indices” The description of the meaning of all formulae used must be provided in this section. For example in the section 2.15.1, the formulae/indices: SSI=1-(Ys/Yp)/SI. What is the meaning of the Ys, Yp and SI? This must be provide in this section and not at the end of the manuscript.

Response: now all the information regarding the meaning of the formulas, abbreviations has been clearly presented for understanding. Please see page 7 and 8 at lines 289-329 

Results:

Comment: 1- For more accuracy, all data should be expressed per leaf dry weight, and not per leaf fresh weight, since there is a change in the leaf relative water content. This will influence the levels of some parameters, such as pigments. Also, the description of the results must follow the statistical data. We can not say that something decreased if it is not significant. This must be carefully revised.

Response: Suggestion is followed and the results have been revised in view of the comments. Please see table 3 and the relevant results. Please see lines 401-425

Comment: 2- The quality of Figures 1 and 2 must be improved. Bars seems to be overlapped.

Response: Now the quality of the figures has been improved. Hope the worthy reviewer will be satisfied with the quality of the figures

Comment: In general the discussion is too long, several parts are repeated, is very confuse and seems more a comparison and presentation of data from other studies. In sum, there is a need for more discussion (not in word count but in quality). How the observed morpho-,physio-, biochemical-, and agronomic changes are related to each other and with the stress imposed? Also the discussion must be supported by the obtained data and not with the available literature.

Response: The whole discussion part is revised clearly in view of the results following the suggestions of the worthy reviewer. Hope the worthy reviewer will be satisfied with the revised discussion part

Comment: Some parts of the discussion are expendable, eg the first 2 paragraphs.

Response: Suggestion is followed  

Comment: The authors failed to give a proper concluding message for the readers.

Response: now the conclusion part is revised with concluding remarks

Round 2

Reviewer 1 Report

Thank you, Authors answered all of my questions.

Author Response

Reviewer 1

Comment: Thank you, Authors answered all of my questions

Response: We are thankful for the satisfaction of the worthy reviewer about the revised manuscript

Reviewer 2 Report

The quality of the document was improved. However,  important points still need correction and improvements. 

  • line 47: 40% and not 40 percent
  • line 308: increase the quality of the formula
  • Concerning my previous comment “For more accuracy, all data should be expressed per leaf dry weight, and not per leaf fresh weight, since there is a change in the leaf relative water content. This will influence the levels of some parameters, such as pigments. ” The authors only presented the data of pigments per dry weight. What about the other parameters? The pigments are only one example. For sure that differences will be found in the other remain parameters expressed in fresh matter (h2O2, SOD, CAT, APX, POD, MDA, AsA, flavonoids and TPC, TSP, anthocyanins, FAA, TSS, RS and NRS). I encourage the authors to proceed with this change to /DW.
  • The Discussion still too long, PLEASE avoid descriptive and well-known facts (this is only one example: line 725 “Under water deficit stress the overproduction of ROS such as hydrogen peroxide (H2O2), singlet oxygen, superoxide, hydroxyl redicals etc. is a common phenomenon. These ROS directly or indirectly damages the cell membranes and other components. Damages to cellular membranes due to overly produced ROS results in the production of MDA due to damages to membrane lipids. The extent of MDA production represents the severity of oxidative stress along with the potential of antioxidative defence [105,106]. This oxidative stress results in the reduction of growth and yield through several disturbances in normal metabolic activities. Plants have evolved a well-developed antioxidative defense mechanism to counteract the damaging effects of oxidative stress. However, the antioxidative defense mechanism is plant species and even cultivar specific [107,108]. It includes the activities of antioxidative enzymes (POD, SOD, APX and CAT) and contents of non-enzymati antioxidants (AsA, phenolics, carotenoids, flavonoids, and anthocyanin). These both types of antioxidantive mechanisms work together in a supportive way [19,106,109-111]. Comprehensively the enzymatic antioxidant cause the conversion of ROS to H2O2 and then to the H2O molecule. SODs catalyze the disproportionation of O2 •¯ into H2O2. While the CAT, APX, and GR catalyze the decomposition of H2O2 into H2O and O2. APX participates in the AsA-GSH (ascorbate-glutathione) pathway as a key enzyme. Non-enzymatic antioxidants detoxify ROS by interrupting a free-radical chain reaction. AsA, compatible solutes, phenollics, alpha-tocopherol, carotenoids, and flavonoids counteract the uncontrolled cascade of ROS produced during stress [112,113]. AsA participates in the AsA-GSH pathway as an electron donor for APX. AsA helps in the regeneration of tocopherol and xanthophyll production that partakes in quenching of the excitation energy. Carotenoids are a light-harvesting pigment alleviates high light illumination induced oxidative stress by quenching excessive energy as heat dissipation. Carotenoids also avert the over-excitation of PS II in the thylakoid membrane by efficiently scavenging 1Chl*, 3Chl*, and 1O2 [111].”). Revise all discussion to avoid long paragraphs with descriptions.

Author Response

Response to Comments

We are thankful for the worthy reviewer about the comments for the improvement of manuscript. The manuscript has been revised in view of the given comments and hope the worthy reviewer will be satisfied with the revision

Comment: line 47: 40% and not 40 percent

Response: Correction has been incorporated in the text

Comment: line 308: increase the quality of the formula

Response: The quality has been improved

Comment: Concerning my previous comment “For more accuracy, all data should be expressed per leaf dry weight, and not per leaf fresh weight, since there is a change in the leaf relative water content. This will influence the levels of some parameters, such as pigments. ” The authors only presented the data of pigments per dry weight. What about the other parameters? The pigments are only one example. For sure that differences will be found in the other remain parameters expressed in fresh matter (h2O2, SOD, CAT, APX, POD, MDA, AsA, flavonoids and TPC, TSP, anthocyanins, FAA, TSS, RS and NRS). I encourage the authors to proceed with this change to /DW.

Response: Suggestion is followed and now all the mentioned attributes (h2O2, SOD, CAT, APX, POD, MDA, AsA, flavonoids and TPC, TSP, anthocyanins, FAA, TSS, RS and NRS) have been presented on dry wt basis. The changes have also been incorporated in results and discussion part. Please see lines 434-438, 449-456, 458-473, 482-484, 488-496, 511-534, and whole the discussion part  

Comment: The Discussion still too long, PLEASE avoid descriptive and well-known facts (this is only one example: line 725 “Under water deficit stress the overproduction of ROS such as hydrogen peroxide (H2O2), singlet oxygen, superoxide, hydroxyl redicals etc. is a common phenomenon. These ROS directly or indirectly damages the cell membranes and other components. Damages to cellular membranes due to overly produced ROS results in the production of MDA due to damages to membrane lipids. The extent of MDA production represents the severity of oxidative stress along with the potential of antioxidative defence [105,106]. This oxidative stress results in the reduction of growth and yield through several disturbances in normal metabolic activities. Plants have evolved a well-developed antioxidative defense mechanism to counteract the damaging effects of oxidative stress. However, the antioxidative defense mechanism is plant species and even cultivar specific [107,108]. It includes the activities of antioxidative enzymes (POD, SOD, APX and CAT) and contents of non-enzymatic antioxidants (AsA, phenolics, carotenoids, flavonoids, and anthocyanin). These both types of antioxidantive mechanisms work together in a supportive way [19,106,109-111]. Comprehensively the enzymatic antioxidant cause the conversion of ROS to H2O2 and then to the H2O molecule. SODs catalyze the disproportionation of O2 •¯ into H2O2. While the CAT, APX, and GR catalyze the decomposition of H2O2 into H2O and O2. APX participates in the AsA-GSH (ascorbate-glutathione) pathway as a key enzyme. Non-enzymatic antioxidants detoxify ROS by interrupting a free-radical chain reaction. AsA, compatible solutes, phenollics, alpha-tocopherol, carotenoids, and flavonoids counteract the uncontrolled cascade of ROS produced during stress [112,113]. AsA participates in the AsA-GSH pathway as an electron donor for APX. AsA helps in the regeneration of tocopherol and xanthophyll production that partakes in quenching of the excitation energy. Carotenoids are a light-harvesting pigment alleviates high light illumination induced oxidative stress by quenching excessive energy as heat dissipation. Carotenoids also avert the over-excitation of PS II in the thylakoid membrane by efficiently scavenging 1Chl*, 3Chl*, and 1O2 [111].”). Revise all discussion to avoid long paragraphs with descriptions.

Response: Thanks for the worthy reviewer for the comment. Now the discussion part has been revised following the comments and has been posssibly reduced by excluding the descriptive part

Round 3

Reviewer 2 Report

The authors answered my question and some parts of the discussion were improved.